# A Bayesian analysis of the association between *Leukotriene A4 Hydrolase* genotype and survival in tuberculous meningitis

Laura Whitworth[1], Jacob Coxon[2], Arjan van Laarhoven[3], Nguyen Thuy Thuong Thuong[4], Sofiati Dian[5,6], Bachti Alisjahbana[5], Ahmad Rizal Ganiem[5,6], Reinout van Crevel[3], Guy E Thwaites[4,7], Mark Troll[1], Paul H Edelstein[1,8], Roger Sewell[2]*, Lalita Ramakrishnan[1]*

[1]Molecular Immunity Unit, Department of Medicine, University of Cambridge, MRC Laboratory of Molecular Biology, Cambridge, United Kingdom; [2]Trinity College, Cambridge, United Kingdom; [3]Department of Internal Medicine and Radboud Center for Infectious Diseases (RCI), Radboud Institute for Molecular Life Sciences (RIMLS), Radboud University Medical Center, Nijmegen, Netherlands; [4]Oxford University Clinical Research Unit, Ho Chi Minh City, Viet Nam; [5]Universitas Padjadjaran, TB-HIV Research Center, Faculty of Medicine, Bandung, Indonesia; [6]Department of Neurology, Faculty of Medicine/Hasan Sadikin Hospital, Universitas Padjadjaran, Sumedang, Indonesia; [7]Centre for Tropical Medicine and Global Health, Nuffield Department of Medicine, University of Oxford, Oxford, United Kingdom; [8]Department of Pathology and Laboratory Medicine, Perelman School of Medicine, University of Pennsylvania, Philadelphia, United States

*For correspondence:
roger.sewell@cantab.net (RS);
lr404@cam.ac.uk (LR)

Competing interests: The authors declare that no competing interests exist.

**Abstract** Tuberculous meningitis has high mortality, linked to excessive inflammation. However, adjunctive anti-inflammatory corticosteroids reduce mortality by only 30%, suggesting that inflammatory pathophysiology causes only a subset of deaths. In Vietnam, the survival benefit of anti-inflammatory corticosteroids was most pronounced in patients with a C/T promoter variant in the leukotriene A$_4$ hydrolase (*LTA4H*) gene encoding an enzyme that regulates inflammatory eicosanoids. *LTA4H* TT patients with increased expression had increased survival, consistent with corticosteroids benefiting individuals with hyper-inflammatory responses. However, an Indonesia study did not find an *LTA4H* TT genotype survival benefit. Here using Bayesian methods to analyse both studies, we find that *LTA4H* TT genotype confers survival benefit that begins early and continues long-term in both populations. This benefit is nullified in the most severe cases with high early mortality. *LTA4H* genotyping together with disease severity assessment may target glucocorticoid therapy to patients most likely to benefit from it.

## Introduction

Tuberculous meningitis (TBM) is the most severe form of tuberculosis. Despite effective antimicrobial therapy, it results in 20–25% mortality in HIV-negative individuals and ~40% mortality in HIV-positive individuals (*Stadelman et al., 2020*; *Thwaites et al., 2013*). A long-standing hypothesis that an excessive intracerebral inflammatory response underlies TBM mortality (*Shane and Riley, 1953*) led to multiple trials of adjunctive anti-inflammatory treatment with corticosteroids (e.g. dexamethasone) (*Prasad et al., 2016*; *Wilkinson et al., 2017*). Findings from a randomised controlled trial (RCT) in

**eLife digest** Tuberculous meningitis is a serious infection of the lining of the brain, which affects over 100,000 people a year. Without treatment, it is always fatal: even with proper antibiotics, about a quarter of patients do not survive and many will have permanent brain damage. Overactive inflammation is thought to contribute to this process. Corticosteroid drugs, which dampen the inflammatory process, are therefore often used during treatment. However, they merely reduce mortality by 30%, suggesting that only some people benefit from them.

Two recent studies have linked the genetic makeup of individuals who have tuberculous meningitis to how they respond to corticosteroids. There were, in particular, differences in the LTA4H gene that codes for an inflammation-causing protein. According to these results, only individuals carrying high-inflammation versions of the LTA4H gene would benefit from the treatment. Yet a third study did not find any effect of the genetic background of patients.

All three papers used frequentist statistics to draw their conclusions, only examining the percentage of people who survived in each group. Yet, this type of analysis can miss important details. It also does not work well when the number of patients is small, or when the effectiveness of a drug varies during the course of an illness.

Another method, called Bayesian statistics, can perform better under these limitations. In particular, it takes into account the probability of an event based on prior knowledge – for instance, that the risk of dying varies smoothly with time.

Here, Whitworth et al. used Bayesian statistics to reanalyse the data from these studies, demonstrating that death rates were correlated with the type of LTA4H gene carried by patients. In particular, corticosteroid treatment worked best for people with the high inflammation versions of the gene. However, regardless of genetic background, corticosteroids were not effective if patients were extremely sick before being treated.

The work by Whitworth et al. demonstrates the importance of using Bayesian statistics to examine the effectiveness of medical treatments. It could help to design better protocols for tuberculous meningitis treatment, tailored to the genetic makeup of patients.

Vietnam that adjunctive dexamethasone improved survival by ~30% led to it becoming standard of care treatment (*Thwaites et al., 2004*). However, the modest benefit of adjunctive dexamethasone treatment suggested a heterogeneity in glucocorticoid-responsiveness (*Donald and Schoeman, 2004*; *Schoeman and Donald, 2013*). Findings in a zebrafish model of TB provided a testable hypothesis for a mechanism underlying this heterogeneity (*Thwaites et al., 2004*; *Tobin et al., 2012*; *Tobin et al., 2010*). The zebrafish findings suggested that either deficiency or excess of leukotriene A4 hydrolase (LTA4H), a key enzyme that regulates the balance of pro- and anti-inflammatory eicosanoids, increase susceptibility to TBM for opposite reasons - too little or too much inflammation (*Tobin et al., 2012*; *Tobin et al., 2010*). It became possible to test the prediction when a common human functional *LTA4H* promoter variant (rs17525495) was identified comprising a C/T transition that controlled LTA4H expression, with the T allele causing increased expression (*Tobin et al., 2012*). A retrospective analysis of patient *LTA4H* rs17525495 genotypes in the Vietnam dexamethasone RCT cohort confirmed the prediction (*Thwaites et al., 2004*; *Tobin et al., 2012*). Among HIV-negative patients, the survival benefit of dexamethasone was restricted to patients with the hyper-inflammatory *TT* genotype, with CC patients potentially harmed by this treatment (*Tobin et al., 2012*). These findings supported the model that mortality from TBM was due to two distinct inflammatory states, and that *LTA4H* genotype might be a critical determinant of inflammation and consequently of the response to adjunctive anti-inflammatory treatment. If true, then personalized genotype-directed adjunctive glucocorticoid treatment would be warranted, with the drug given only to TT patients. This would be particularly important given the possible harm to the hypo-inflammatory CC group, as well as the adverse effects of long-term high dose treatment with a broadly acting immunosuppressant.

To further these findings, two new studies of the association of *LTA4H* genotype with TBM survival in HIV-negative patients were performed in Vietnam and Indonesia, respectively (*Thuong et al., 2017*; *van Laarhoven et al., 2017*). Because glucocorticoid adjunctive therapy had become

## Box 1. Contrast of '95% significant' in Bayesian and frequentist paradigms (*MacKay, 2003*).

Bayes: 'A is significantly greater than B' = Posterior probability that A greater than B is at least 0.95.

Frequentist: 'A is significantly greater than B' = For any circumstance where A is at most B, the probability of getting data in this critical region, as we did, was at most 0.05.

Therefore:

1. We expect 1 in 20 of Bayesianly (95%) significant results to be truly negative and therefore false positives;

2. We expect up to 1 in 20 of truly negative results to be frequentistly significant (at the 95% level) and therefore false positives.

Therefore (assuming all positives are at 95% level):

- In the frequentist paradigm, the expected number of false positive results is proportional to the number of comparisons done on true negatives;

- In the Bayesian paradigm, the expected number of false positive results is proportional to the number of apparent positive results, and unaffected by any vast number of accompanying apparent negative results.

**Table 1.** Characteristics of the Vietnam and Indonesia cohorts.

Bayesian posterior probabilities comparing the two cohorts are shown (probability that mean of starred group is higher, ** > 0.99, *** > 0.999, all other comparisons, not significant). See also Figure S1 for probability differences for each GCS.

|  | Vietnam | Indonesia |
|---|---|---|
| Total | 439 | 376 |
| **Glasgow Coma Score** | | |
| *mean* | 13.3 ** | 12.8 |
| *(range)* | (3-15) | (5-15) |
| **BMRC TBM grade** | | |
| *no. (% of total)* | | |
| 1 | 163 (37.1) *** | 34 (9.0) |
| 2 | 206 (47.0) | 284 (75.5) *** |
| 3 | 70 (15.9) | 58 (15.4) |
| **Age in years** | | |
| *median* | 41 *** | 28 |
| *(range)* | (18-93) | (14-90) |
| **Age in years by TBM grade** | | |
| *median (range)* | | |
| 1 | 39 (18-85) *** | 27 (16-45) |
| 2 | 47 (18-93) *** | 29 (14-90) |
| 3 | 33 (18-86) *** | 26 (14-64) |
| **Overall mortality** | | |
| *no. (%)* | 83 (18.8) | 146 (39.9) *** |
| **Time to median mortality** | | |
| *(days)* | 50 *** | 8 |
| **Mortality by BMRC TBM grade** | | |
| *No. dead (% of grade)* | | |
| 1 | 12 (6.8) | 5 (15.9) |
| 2 | 45 (21.9) | 106 (38.0) *** |
| 3 | 26 (37.9) | 35 (63.8) ** |

## Box 2. TBM disease severity classification.

*Glasgow Coma Score (GCS)* A general measure of consciousness used for a wide range of neurological deficits, particularly brain trauma, by scoring eye opening and verbal and motor responses to stimuli, to assign a numerical value from 3 to 15 corresponding to decreasing severity, where 3 corresponds to completely unresponsive, deep coma and 15 to fully conscious (*Teasdale and Jennett, 1974*).

*Modified British Medical Research Council (BMRC) TBM Grade* A classification scheme specifically tailored to assess TBM severity. It is derived from the GCS, and additionally incorporates the presence of focal neurological signs. The TBM grade is scored between 1–3 corresponding to increasing severity, converse to the GCS classification.

*GCS and BMRC Grade relationship* BMRC Grade 1 - GCS = 15 with no focal neurological signs; BMRC Grade 2 - GCS = 11–14, or GCS = 15 with focal neurological signs, BMRC Grade 3 - GCS < 10 (*Heemskerk et al., 2011*).

standard of care owing to the benefit observed in the randomised controlled trial (*Thwaites et al., 2004*), all patients received it in both studies. Therefore, the prediction that could be tested was that TT mortality is less than CC+CT mortality. Whereas the Vietnam study confirmed this prediction, the Indonesia study did not. The Vietnam cohort had an overall mortality of 18.8%, similar to that reported in the literature (*Stadelman et al., 2020*). A striking feature of the Indonesia cohort was its more than two-fold increased mortality in comparison with the Vietnam cohort. Moreover, most of the Indonesia cohort deaths occurred early with a median time to death of eight days versus 50 days in Vietnam (*Table 1*). This high early mortality raised the possibility that the impact of the *LTA4H* variant differs by disease severity, and may not be relevant in more severe disease (*Fava and Schurr, 2017*). If so, then the effects of the *LTA4H* genotype were being masked by the preponderance of extremely severe cases in the Indonesia cohort (*Fava and Schurr, 2017*).

Both studies used, as the primary metric of significance testing, Cox regression modelling, an approach that assumes that the ratio of hazard rates between groups is constant throughout the observed period (*Bradburn et al., 2003*; *Greenland et al., 2016*). Therefore, this analytical method could miss important differences in these studies of TBM, a disease which can present acutely yet have a prolonged time-course with vastly differing mortality risks over time (*Thuong et al., 2017*; *Thwaites et al., 2013*; *van Laarhoven et al., 2017*). Moreover, testing the hypothesis that *LTA4H* effects are limited to specific disease severity grades requires subgroup analysis. The use of frequentist statistics would limit the ability to perform such subgroup analyses because the penalties it sets for multiple comparisons do not reflect real-world situations (*Gelman and Loken, 2013*; *Smith and Ebrahim, 2002*). Bayesian analysis is ideally suited to simultaneously estimate treatment effects in multiple subgroups because it results in different interactions with the number of results obtained which are much less problematic than those arising in frequentist analysis (*Box 1* and Appendix 1).

Therefore, we used a Bayesian approach to analyse data from the two cohorts (*Gelman and Loken, 2013*; *MacKay, 2003*; *Zampieri et al., 2020*) (Appendix 2). Bayesian analysis also enables the detection of significant differences that might be limited to just a part of the time-course and therefore would allow analysis to be independent of the kinetics of death in the Vietnam and Indonesia cohorts. Medical management decisions are guided by an assessment of the probabilities of outcome. In TBM, the question faced by the clinician is how likely is glucocorticoid therapy going to help or harm a patient. Bayesian paradigms, unlike frequentist ones, understand probability in a real-world way, using it to indicate the plausibility of a particular conclusion (*MacKay, 2003*; *Zampieri et al., 2020*). Finally, relevant to this re-analysis of completed clinical studies, Bayesian paradigms have less potential for bias arising from post-hoc analysis (Appendix 1).

The severity grade-specific analyses, coupled with temporal analyses made possible by Bayesian methods, reveal that the *LTA4H* TT genotype is associated with survival in both cohorts.

## Results

### Characteristics of the Vietnam and Indonesia cohorts

The age range of patients was similar in the Vietnam and Indonesia cohorts with Indonesia patients tending to be younger (*Table 1*). We compared the cohorts for disease severity on presentation using both measures used in the studies, the Glasgow Coma Score (GCS) and the modified British Medical Research Council TBM grade (TBM grade) (*Box 2*). The Indonesia cohort had more severe disease on presentation by both measures (*Table 1* and *Figure 1*). We used the TBM grade for further analyses because it divides patients into just three severity groups, making comparisons more feasible. Importantly, it also provides clinically relevant separation of GCS 15 patients, the most highly represented in both cohorts (*Figure 1*), into those with and without focal neurological signs.

### LTA4H TT genotype association with survival becomes stronger with increasing disease severity in Vietnam

Because the Indonesia cohort was skewed towards more severe disease on presentation, one explanation for the lack of an *LTA4H* genotype association with survival in Indonesia was that the underlying association is overridden by severe disease, a strong independent correlate of mortality (*Fava and Schurr, 2017*; *Wang et al., 2019*). Indeed, a detailed comparison of the Vietnam and Indonesia cohorts showed that 76% of Indonesia patients presented in Grade 2 versus 47% of the Vietnam patients (*Table 1*). This increase was driven entirely by a shift from Grade 1 (9% vs 37% in Vietnam). The cohorts had nearly equal proportions of Grade 3 patients (15% each). Therefore, the ~2 fold-increased overall mortality in Indonesia could be largely accounted for by a corresponding increase of Grade 2 patients (1.6 fold higher than Vietnam). Disease severity as assessed by BMRC Grade or GCS at presentation is also a strong predictor of earlier death (*Fava and Schurr,*

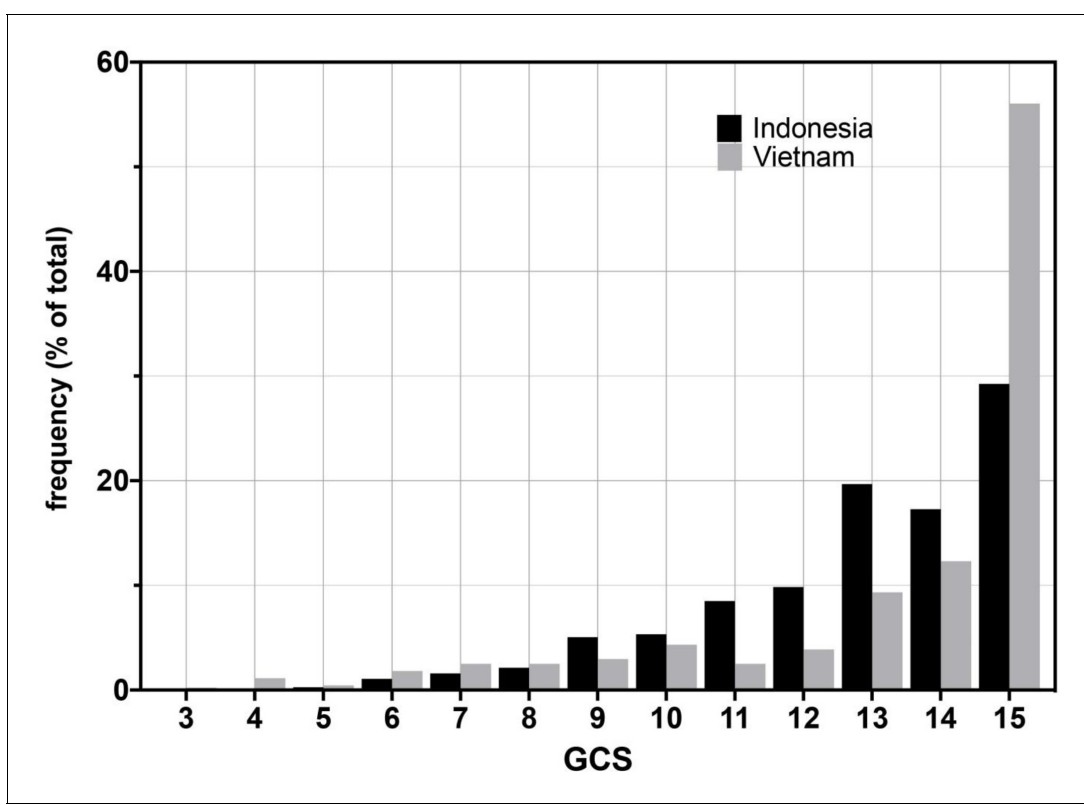

**Figure 1.** Glasgow Coma Scores (GCS) for Vietnam and Indonesia patients. Frequency of GCS values indicated on the Y-axis as a percentage of the total cohort (n = 376 Indonesia, n = 439 Vietnam). Bayesian posterior probabilities of significant differences between Vietnam (VN) and Indonesia (IN) for mean GCS comprising Grade 2, VN > IN P 0.99996 (15 VN > IN p=0.99999; 11–14 VN < IN P ranging from 0.99985 (13) to 0.98 (14); rest of the values non-significant); for GCS comprising Grade 3, VN > IN, P 0.01 (GCS4, 0.98; GCS9, 0.04; all others not significant).

*2017*; *Wang et al., 2019*), and the Indonesia patients died sooner (median time to death 8 days versus 50 days in Vietnam) (*Table 1*).

If *LTA4H* genotype associates with survival most strongly in mild disease, then the association seen for the entire Vietnam cohort (*Thuong et al., 2017*) should be strongest in Grade 1 patients. We tested this prediction with Bayesian analysis using a prior that was intentionally uninformative and very wide, while still being centred on clinician-expected survival curves and hazard rates. We included additional parameters to allow for the possibility that not all patient deaths would be linked to the same mode of death. Importantly, the model and priors used allowed us to incorporate our pre-existing knowledge that mortality risk to a population of TBM patients varies smoothly with time, rather than occurring at a number of discrete times common to all patients as is implied by the maximum likelihood solution illustrated by a Kaplan-Meier plot. The details of the model and the priors are in Appendix 2. The definitions of terms and abbreviations used throughout the paper are in *Box 3*.

In the original Vietnam study, the TT genotype was associated with survival and the CC and CT genotypes had similarly increased mortality over TT (*Thuong et al., 2017*), so we compared TT survival to that of CT and CC combined (non-TT). We first confirmed that the TT variant distribution did not differ by grade on presentation (*Table 2*). Bayesian analysis confirmed that the TT genotype was associated with a survival advantage in the overall cohort (*Figure 2A*). Moreover, the analysis revealed that this survival advantage manifested early and persisted over most of the observation period (*Figure 2A*). The Bayesian method enables a more detailed evaluation of mortality risk over time through a hazard rate analysis. This analysis reinforced the significantly higher hazard rate for non-TT starting at 4 days and persisting through 120 days (*Figure 2A*, inset). As predicted by the earlier analysis, the survival probability between CC and CT was not significantly different at any time during the 270 day observation period (data not shown).

When we stratified the Vietnam patients by grade and *LTA4H* genotype, we got a surprising result. The *LTA4H* TT association with survival was barely present in Grade 1, a bit more in Grade 2, and strongest in Grade 3 where it reached significance (*Figure 2B–D*). Similar to the overall cohort, the Grade 3 increased survival probability for TT also manifested early and persisted throughout (*Figure 2D*). Hazard rate analysis again showed that non-TT patients had a greatly increased risk of mortality very early (*Figure 2D*, inset). The non-TT over TT hazard rate ratio peaked at 16 on day 3 (*Figure 2D*, inset). This early high peak dropped only gradually over time; it was 5 at day 100 and remained >1 throughout (data not shown). Again survival of CC and CT patients was not significantly different at any time during the 270 day observation period (data not shown).

In sum, our analysis revealed that in Vietnam, *LTA4H* TT was associated with survival, not in mild disease as suggested earlier (*Fava and Schurr, 2017*), but rather in the most severe disease grade. In fact, the bulk of the overall association was being driven by Grade 3 patients who constituted only 15.9% of the cohort (*Table 1*). Non-TT patients were at greatest risk of dying within days of admission, a risk that diminished with time but remained greater than the TT patients throughout.

## In Indonesia, the LTA4H TT genotype effect does not extend beyond Grade 2

Bayesian analysis found that, in the overall Indonesia cohort, survival of the TT patient group was higher than non-TT though falling short of significance (maximum probability 0.92) (*Figure 2E*). Moreover, this analysis detected that the hazard rate for non-TT patients was higher than TT patients from day 1 to day 13; the non-TT over TT ratio reached significance on days 2 and 3, at which time the non-TT hazard rate was twice that of TT. Thus, while the TT beneficial effect was weaker than in Vietnam (compare *Figure 2E–A*), hazard rate analysis showed that as in Vietnam, TT benefit manifested early (*Figure 2E* inset, compare to *Figure 2A* inset). For the grade-stratified cohorts, the analysis of Grade 1 patients was uninformative as there was only one TT patient in this group who survived throughout (*Table 2* and *Figure 2F*; also see Appendix 2, section 4). In Grade 2, the TT survival effect was significant, in contrast to Grade 2 Vietnam (compare *Figure 2G–C*). Rather, the pattern of the Grade 2 association was similar to Vietnam Grade 3 with a significant early TT survival benefit. As in Vietnam, the TT survival benefit started within days with an early hazard rate peak for the non-TT group. In Grade 3, the *LTA4H* TT effect was again absent (*Figure 2H*). As in Vietnam, survival of CC and CT patients was not significantly different at any time during the analysis for the overall cohort and for each of the three grades (data not shown).

## Box 3. Definitions and usages.

Definitions.

- *Posterior probability* - the probability after seeing the data
- *Mean posterior survival probability at time T* - the expectation after seeing the data of the fraction of patients that will still be alive at time T
- *Hazard rate* - the fraction of those still surviving that will die per unit time. A high hazard rate at a particular time indicates that patients are at high risk of dying at that time
- *Mean posterior hazard rate at time T* - the expectation after seeing the data of the hazard rate at time T

Abbreviated and example usages.

- *Onwards* - for the rest of the 270 day observation period.
- *Throughout* - for the entire 270 day observation period.
- *'A was significantly greater than B at time T'* - the posterior probability that A was greater than B, at time T, was at least 0.95.
- *'A was not significantly different from B'* - the posterior probability that A was greater than B was between 0.05 and 0.95 throughout.
- *'Group A survival was 30% greater than group B'* - the mean posterior survival probability at 270 days, $p_A$, was 30% absolute greater than the corresponding probability $p_B$ for group B. (*'absolute'* here meaning that $p_A = p_B + 0.3$, and not that $p_A = p_B \times 1.3$).
- *'Probability that group A survival was better than group B at time T was 0.97'* - the posterior probability that group A survival probability at time T was greater than group B survival probability at time T was 0.97 (Note that this is not a reference to the mean posterior survival probability).
- *'The hazard rate ratio for group A over group B peaked at X at time T and remained greater than Y throughout'* or *'Group A had an X-fold higher relative risk of death at time T which remained greater than Y throughout'* - the mean posterior hazard rate for group A, divided by that for group B, peaked at a value of X at time T and remained greater than Y at all times up to 270 days.
- *'The probability that hazard rate for group A was greater than that of group B was 0.97 at time T'* - the posterior probability was 0.97 that the hazard rate for group A at time T was greater than that for group B at time T (Note that this is not a reference to the mean posterior hazard rate).
- *'The hazard rate for group A was greater than that for group B at time T'* - the mean posterior hazard rate for group A was greater than that for group B at time T.
- *'survival gap'* - the difference in mean posterior survival probability between the two groups being considered at 270 days.

Since Grade 2 patients constitute the bulk of the Indonesia cohort (75.5%), why was the *LTA4H* genotype effect in this grade not reflected in the overall cohort analysis? This was particularly curious given that in Vietnam the overall significant effect was being driven very substantially by Grade 3 patients who constituted only 15.9% of the cohort. We saw that the *LTA4H* TT benefit was weaker and less prolonged in Indonesia Grade 2 than in Vietnam Grade 3 (compare *Figure 2G to D*). Non-TT patients had similar mortality in Indonesia Grade 2 and Vietnam Grade 3 (compare *Figure 2G to D*).

Thus, Bayesian analysis revealed an *LTA4H* TT survival association in Indonesia as well. The association being only in Grade 2 and not in Grade 3 patients suggested an upper limit of disease severity for its efficacy.

**Table 2.** *LTA4H* genotype frequency in Vietnam and Indonesia.

Bayesian posterior probabilities comparing the two cohorts (probability that starred group is higher, * > 0.95, ** > 0.99, *** > 0.999, all other comparisons, not significant). Comparisons within each cohort yielded no significant differences in *LTA4H* genotype frequencies by grade.

|  | Vietnam | Indonesia |
|---|---|---|
| **rs17525495 *LTA4H*** | | |
| **no. (% total)** | | |
| CC | 184 (41.9) | 216 (57.5)*** |
| CT | 212 (48.3)*** | 128 (34.0) |
| TT | 43 (9.8) | 32 (8.5) |
| **No. (% of grade total)** | | |
| **Grade 1** | | |
| CC | 64/163 (39.3) | 21/34 (61.8)* |
| CT | 81/163 (49.7) | 12/34 (35.3) |
| TT | 18/163 (11.0) | 1/34 (2.9) |
| **Grade 2** | | |
| CC | 86/206 (41.8) | 161/284 (56.7)*** |
| CT | 100/206 (48.5)*** | 99/284 (34.9) |
| TT | 20/206 (9.7) | 24/284 (8.5) |
| **Grade 3** | | |
| CC | 34/70 (48.6) | 34/58 (58.6) |
| CT | 31/70 (44.3) | 17/58 (29.3) |
| TT | 5/70 (7.1) | 7/58 (12.1) |

## Grade-for-grade mortality rate differences may explain the difference in grade-specific LTA4H TT effects in Vietnam and Indonesia

Why might *LTA4H* effects stop after Grade 2 in Indonesia? A closer comparison of the overall survival between the two sites suggested that there were mortality differences between the two cohorts for all grades combined and in grade-for-grade comparisons that were *LTA4H*-independent (*Figure 2*). For instance, Indonesia Grade 2 non-TT patients had a mortality risk similar to that of Vietnam Grade 3 non-TT patients (compare *Figures 2D* and *1G*).

We confirmed by Bayesian analysis that within each cohort, mortality risk increased with grade severity (*Thuong et al., 2017*; *van Laarhoven et al., 2017*; *Figure 3*). From early on, Grade 1 survival was significantly greater than Grade 2, which was significantly greater than Grade 3 (*Figure 3A and B*). The hazard rate ratios highlighted that while the risk of increased mortality with higher grade was highest early, it was sustained long-term (*Figure 3C and D*). Importantly, both the survival and hazard rate analyses again pointed to an increased grade-for-grade risk of mortality in Indonesia over Vietnam. Next, we directly compared grade-for-grade mortality between the cohorts. Vietnam survival probability was higher in all grades, and significantly so for Grades 2 and 3 with survival gaps of 18% and 24%, respectively (*Figure 4A,C and E*). Similarly, hazard rates did not differ significantly between the cohorts in Grade 1 (*Figure 4B*), but there was a significant and substantial increase in early hazard rates for Indonesia Grades 2 and 3 (*Figure 4D and F*). We ruled out the possibility that this increased mortality was because the Indonesia patients had more severe disease within each grade as reflected in the GCS. While they had somewhat more severe disease in Grade 2, in Grade 3 where the difference was the most pronounced, they had milder disease (*Figure 1*).

In sum, these analyses show that the inherent higher mortality associated with more severe disease on presentation was sharply accentuated in Indonesia. Indonesia grade 2 patients experienced similar mortality risk as Vietnam Grade 3 patients with the Indonesia Grade 3 patients experiencing far greater mortality. This higher mortality could potentially explain the finding that the *LTA4H* TT survival advantage did not extend to Indonesia Grade 3 patients. It may be that the TT genotype advantage, in response to corticosteroid treatment, is overridden by other factors that cause extreme mortality.

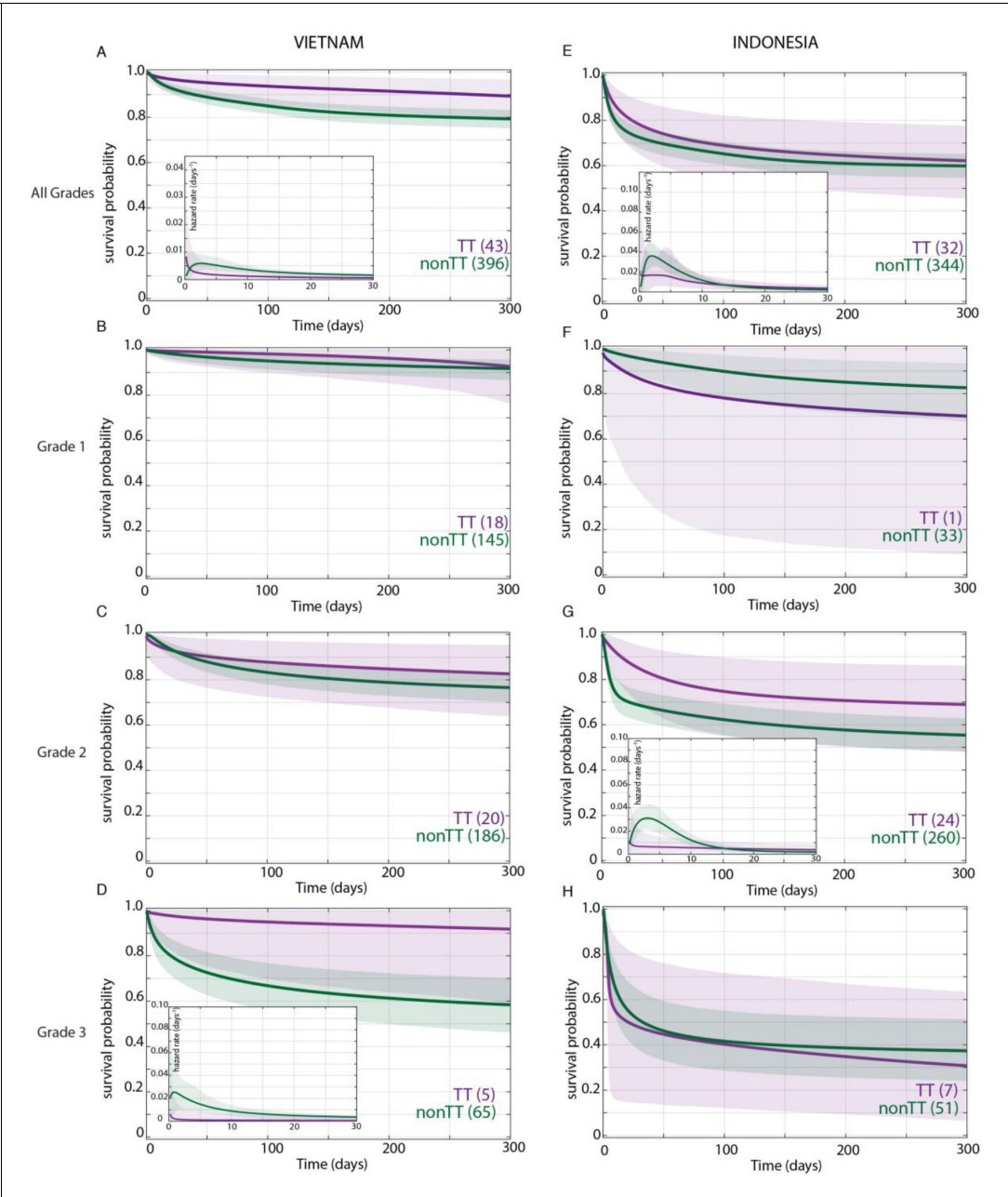

**Figure 2.** Effect of *LTA4H* rs17525495 genotype on patient survival. Survival probability over all grades in Vietnam (**A**) and Indonesia (**E**), and stratified by grade (**B–D, F–H**). Coloured lines represent mean posterior survival probability curves for the nine-month observation period. Shaded areas represent 95% Bayesian confidence limits for posterior probability. Comparisons where TT (purple) to non-TT (green) differences were significant have boxed insets showing hazard rates for the first 30 days; all other comparisons, not significant. The number of patients at the starting time point are indicated in parentheses. In Vietnam, overall (**A**), TT survival was significantly higher than non-TT from day 39 onwards with maximum probability 0.98, survival gap 11%; non-TT hazard rate was significantly higher than TT from day 4 to day 120, with their ratio peaking at 3 on day six and remaining >1 until day 223. (**D**) Grade 3 TT survival was significantly higher from day 3 onwards with maximum probability 0.97, survival gap 30%. The TT hazard rate dropped from the start, while the non-TT hazard peaked at 16 times higher than TT on day 3; non-TT over TT hazard rate ratio remained >1 throughout. In Indonesia, overall (**E**), TT survival was non-significantly higher than non-TT (maximum probability 0.92); the non-TT hazard rate was greater than the TT hazard rate from day 1 to day 13, significantly so (and by 2-fold) on days 2 and 3 (maximum probability 0.97). (**F**) Grade 1 comparisons were uninformative due to TT sample size (n = 1). (**G**) Grade 2 TT survival was significantly higher on days 4–32 with maximum probability 0.99, survival gap 9%. The TT hazard rate dropped from the start, while the non-TT hazard peaked at five times higher than TT on day 3. The non-TT over TT hazard rate ratio remained >1 until day 15.

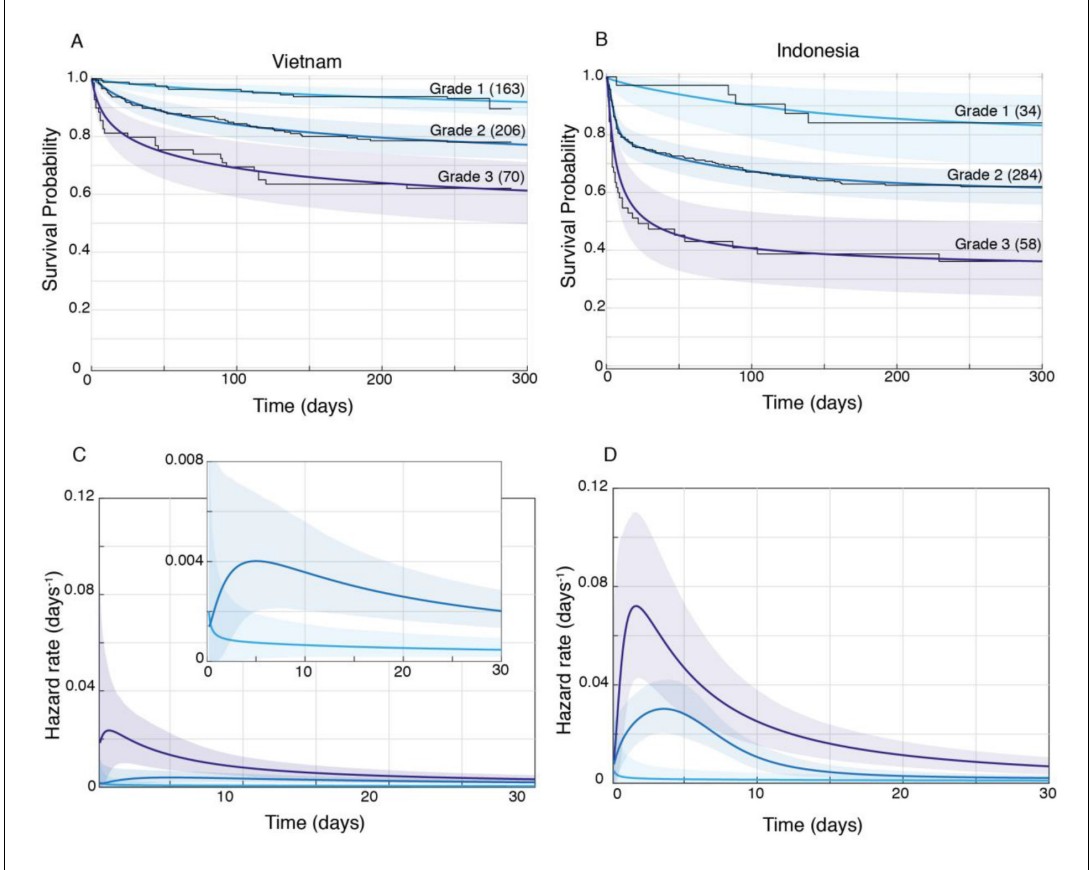

**Figure 3.** Patient survival, stratified by grade. Mean posterior survival probability curves (coloured lines) overlaid by Kaplan-Meier survival plots (black lines) for Vietnam (**A**) and Indonesia (**B**), and mean posterior hazard rate curves for the first 30 days for Vietnam (**C**) and Indonesia (**D**). Shaded areas represent the 95% Bayesian confidence limits for posterior probability. The number of patients in each group at the starting time point are indicated in parentheses. (**A**) Vietnam Grade 1 over Grade 2 survival was significantly greater from day 7 onwards with maximum probability 0.999, survival gap 14%; Grade 2 over Grade 3 survival was significantly greater from day 1 onwards with maximum probability 0.999, survival gap 16%. (**B**) Indonesia Grade 1 over Grade 2 survival was significantly greater from day 2 onwards with maximum probability 0.999, survival gap 21%; Grade 2 over Grade 3 survival was significantly greater from day 2 onwards with maximum probability 0.999, survival gap 25%. (**C**) Vietnam hazard rate ratio was >1 for both grade comparisons (inset magnifies Grade 1 and Grade 2 differences) with Grade 2 over 1 ratio peaking at 5.5 on day 7 and Grade 3 over 2 ratio peaking at 13.9 on day 1. (**D**) Indonesia Grade 2 over 1 hazard rate ratio was >1 up to day 215 and Grade 3 over 2 ratio was >1 throughout, peaking at 11.2 on day 1 for Grade 2 over 1, and at 2.3 on day 5 for Grades 3 over 2.

## Discussion

The finding in two independent cohorts in Vietnam collected from 2001 to 2004 and 2011–2015 that a common functional human variant was associated with responsiveness to adjunctive glucocorticoid treatment in TBM represented an ideal example of pharmacogenomics, coming as it did from mechanistic understanding of the underlying reason (*Sadée and Dai, 2005*; *Thuong et al., 2017*; *Tobin et al., 2012*). The potential importance of these findings was heightened with subsequent pathway analyses in the zebrafish (where the *LTA4H* link to disease severity was discovered) identifying more specific, inexpensive, steroid-sparing drugs that might circumvent LTA4H-mediated pathology (*Roca and Ramakrishnan, 2013*; *Roca et al., 2019*). It was disappointing and puzzling when a similar study of TBM patients in Indonesia failed to find a significant association (*van Laarhoven et al., 2017*). In a commentary published alongside the Indonesia and second Vietnam studies, Fava and Schurr suggested that background genetic differences between the cohorts might account for the lack of an *LTA4H* association (*Fava and Schurr, 2017*). However, noting that the Indonesia cohort presented with more severe disease and died earlier, they postulated that rather than invoking an unknown genetic phenomenon, it was more likely that the beneficial effects of dexamethasone for the *LTA4H* TT genotype were nullified in more severe disease. This concept of the disease

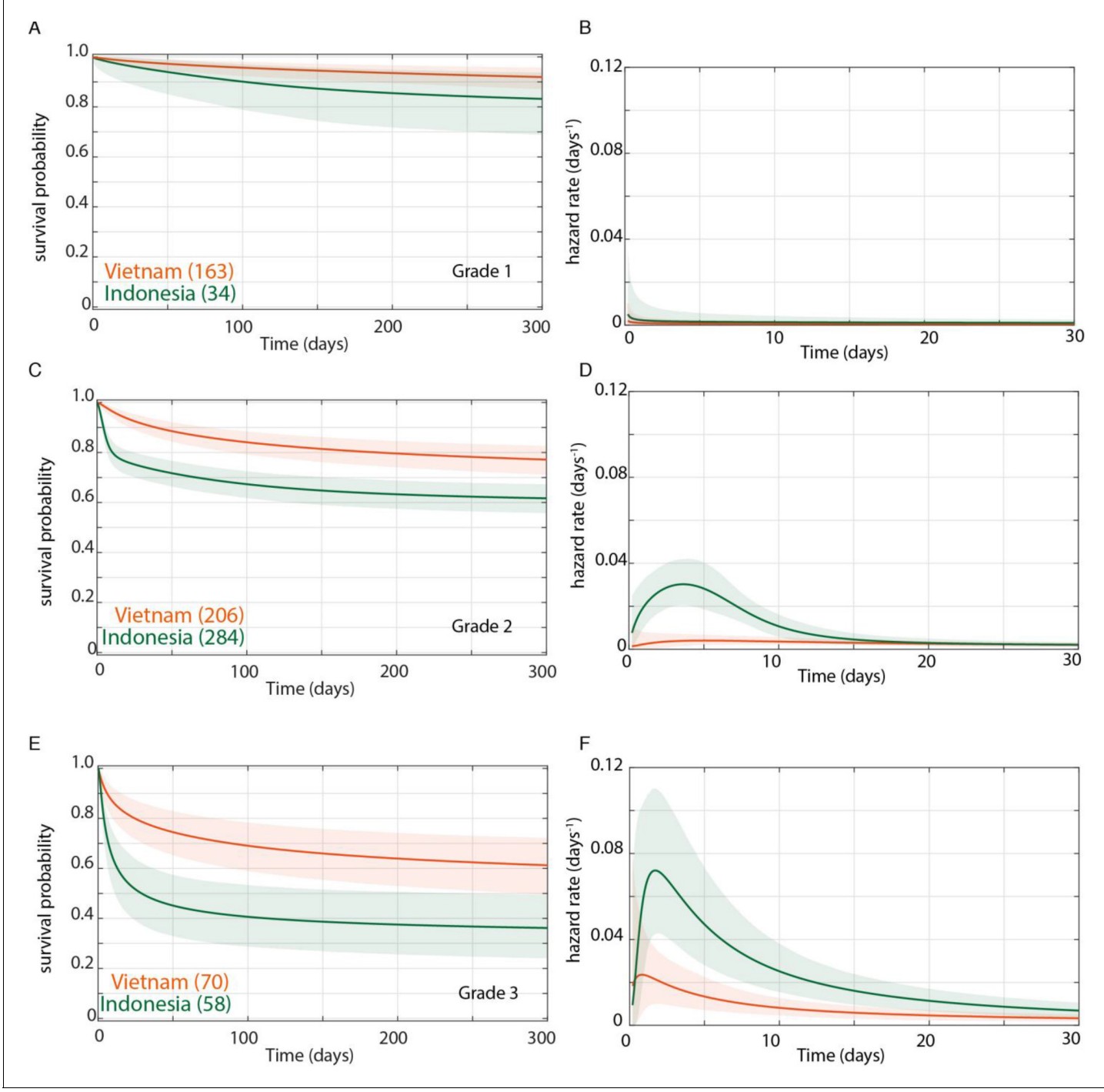

**Figure 4.** Direct comparison of grade-stratified survival of Vietnam and Indonesia patients. Comparison of survival curves (A, C, E) and hazard rate curves for the first 30 days (B, D, F) in Vietnam (orange lines) and Indonesia (green lines). The number of patients at the starting time point are indicated in parentheses. (A) Grade 1 survival did not differ significantly between Vietnam and Indonesia. (B) Indonesia hazard rate was ~2 fold higher than Vietnam in Grade 1 from day 0 to day 150, but the difference did not reach significance. (C) Grade 2 survival was significantly lower in Indonesia than Vietnam from day 1 onwards with maximum probability 0.999, survival gap 18%. (D) The hazard rate ratio in Grade 2 for Indonesia over Vietnam peaked at 8.4 on day 1 and remained >1 until day 180. (E) Grade 3 survival was significantly lower in Indonesia than Vietnam from day 2 onwards with maximum probability 0.999, survival gap 24%. (F) Grade 3 hazard rate ratio for Indonesia over Vietnam peaked at 3.6 on day 3 and remained >1 until day 165.

being too severe for outcomes to be influenced by intervention has precedent. The beneficial effect of fluoroquinolones in TBM is present in Grade 1 or 2 disease, but lost in Grade 3 disease (*Thwaites et al., 2011*). To test if increased grade severity on presentation in Indonesia was sufficient to account for the loss of the *LTA4H* TT survival effect, a grade-for-grade analysis within and between the two cohorts was necessary, while taking into account the temporal changes in mortality risk over the several months-long observation period. Moreover, both the magnitude and the time to mortality seemed to vary dramatically between the cohorts (*Table 1*). We realized that Bayesian methods were ideal for these complex and intrinsically multivariate comparisons and possibly the only path to a biologically and clinically relevant understanding.

When we analysed the Vietnam cohort separated by grade severity, we saw that there was indeed a relationship between *LTA4H* effect and grade severity. However, it was opposite to what had been predicted (*Fava and Schurr, 2017*). The TT survival benefit became stronger not weaker with increasing grade. The analysis also suggested the reason for this. Patients with mild disease on presentation did well regardless of *LTA4H* genotype, so that the added benefit of the TT genotype was small. In Indonesia, the *LTA4H* TT effect was present in Grade 2 and completely absent in Grade 3, a finding that was perplexing until we analysed the *LTA4H*-independent mortality risk of the two cohorts. The Indonesia cohort did not just contain a greater number of patients presenting at a more severe grade as had been noted earlier (*Fava and Schurr, 2017*), but patients in this cohort had substantially higher early mortality than their Vietnam counterparts even grade-for-grade (*Table 1*). Indonesia patients had nearly twice the mortality risk in Grade 2 and ~50% higher risk in Grade 3. Tellingly, Grade 2 Indonesia overall mortality risk was virtually identical to Grade 3 Vietnam (38% versus 37.9%), suggesting that this level of overall mortality represents the boundary of the beneficial effect of *LTA4H* TT. Thus, *LTA4H* TT efficacy was limited by other factors that cause mortality. These factors appear independent of severity grade on presentation, and if they exceed a threshold (represented by about ~40% mortality) then the beneficial effect of *LTA4H* TT is lost. A limitation of this study is that we cannot definitively determine the causes of this *LTA4H* genotype-independent excess Grade 3 mortality in Indonesia because the patient populations, hospital care protocols and the availability of intensive care differed between each study site. One possibility is that better ancillary care was possible in Vietnam where all patients were enrolled into a clinical trial versus only 17% in Indonesia (*Thuong et al., 2017*; *van Laarhoven et al., 2017*). The excess Grade 3 Indonesia mortality peaked within days, and optimized respiratory support, in particular, would be essential to keep patients alive through the early high risk stage in order allow for anti-inflammatory effects of corticosteroids to benefit the TT patients. Thus the beneficial effect of dexamethasone to the *LTA4H* TT group may simply not have had time to come into play in Grade 3 Indonesia patients. A second possibility is that Indonesia Grade 3 patients presented with a greater degree of dysregulated inflammation in a manner not revealed by standard metrics of judging disease severity. If this were the case then corticosteroids might no longer be beneficial. Since both TT and non-TT patients suffered identical excess mortality risk, its cause would have been *LTA4H*-independent. Indonesia patients tended to be younger than Vietnam patients in all grades (*Table 1*), and perhaps were more prone to develop such a response. Such a previously unrecognized form of dysregulated inflammation might be caused by other genetic variants uniquely present in the Indonesia patients and would work through an inflammatory pathway that is less responsive to dexamethasone.

Why was the *LTA4H* TT effect thought to be limited to the least severe patients in both cohorts (*Fava and Schurr, 2017*)? Two reasons might explain this. First, it was not appreciated that apart from having fewer Grade 1 patients, Indonesia patients also suffered higher grade-for-grade mortality risk (*Fava and Schurr, 2017*). Second, because of a paucity of less severe patients, a subgroup analysis was performed in an attempt to tease out an effect in this group. Patients were divided into GCS 14–15 (less severe) versus ≤13 (more severe), and a nonsignificant TT survival benefit was found in the GCS 14–15 group only, supporting the idea that TT effects, if any, were limited to the less severe group (*Figure 5A and B*). However, our analysis now shows the two reasons why this grouping resulted in the *LTA4H* TT effect going unrecognized: (1) GCS 15 does not separate patients by whether or not they have neurological signs, so it includes all BMRC Grade 1 and some BMRC Grade 2 patients (*Box 2*); (2) the grouping used resulted in the Grade 2 patients being split so that some were grouped with Grade 3 and the remaining with Grade 1; (3) of the 11 TT patients in the GCS 14–15 grouping only 1 TT patient was in Grade 1, with the remaining 10 in Grade 2, so the effect seen in their GCS 14–15 was entirely from Grade 2 patients (compare *Figure 5B–C*). In any case this

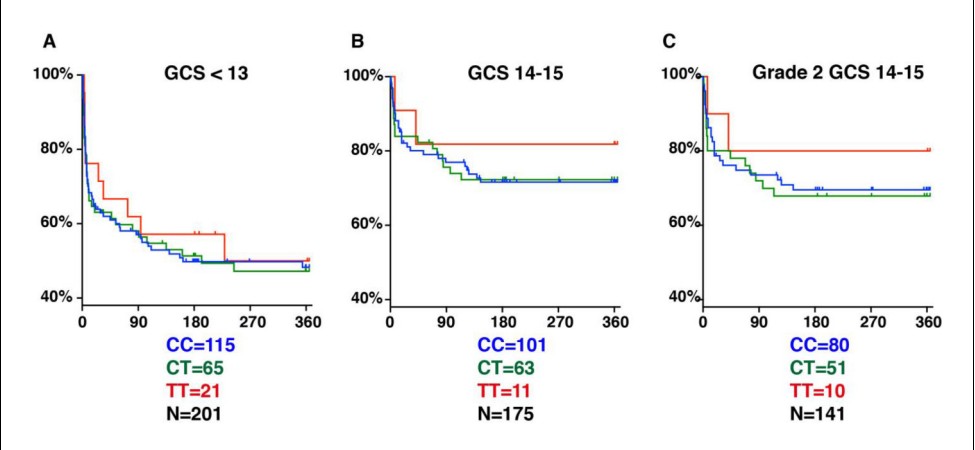

**Figure 5.** *LTA4H* rs17525495 genotype as predictor for 365 day mortality in HIV-negative TBM patients in Indonesia. Panels A and B are comparable to Figure S2B of *van Laarhoven et al., 2017*. (**A**) All patients with GCS < 13. (**B**) Patients with GCS = 14 or 15 (note that one patient with TT genotype, GCS = 15 was censored on day 30 in the original data set but later found to have died on day 41). (**C**) The subset of patients from (B) with GCS 14 or 15 excluding those in BMRC Grade 1 (GCS 15 without neurological signs).

subgroup TT effect would have been impossible to detect with the frequentist analyses used. The use of Bayesian methods has identified subgroups of TT patients that gain the most survival benefit from dexamethasone treatment. Our findings set the stage for using Bayesian methods for subgroup analyses of the ongoing trial with the goal of comparing the survival of CC and CT patients randomised to dexamethasone or placebo, and also comparing the survival of each of these groups individually and combined to TT patients who are all getting dexamethasone (*Donovan et al., 2018*; *Donovan and Phu, 2020*).

Finally, in addition to providing guidance for TBM pharmacogenomic approaches, we hope that our analyses highlight the unique value of Bayesian methods for providing guidance for other complex diseases with difficult treatment decisions. The vital importance of defining the patient populations and subgroups which will benefit the most from specialised interventions and treatments is increasingly appreciated (*Sadée and Dai, 2005*). This is not only to target such treatments to those who will benefit, but to avoid their adverse effects in those individuals who have little chance of experiencing a clinically relevant benefit from them. Our finding that the *LTA4H* TT genotype's salutary role is incumbent on the optimization of other factors that maximize patient survival has broad implications for pharmacogenomic approaches.

## Materials and methods

The anonymized patient cohort data used here has been previously described in detail (*Thuong et al., 2017*; *van Laarhoven et al., 2017*). Our analysis included all 439 HIV-negative patients from the Vietnam study and 376 of the 515 HIV-negative patients from the Indonesia study. The remaining 139 Indonesian patients were excluded for lack of information on *LTA4H* rs17525495 genotype (87), baseline disease severity (44) or outcome (8) (*Supplementary file 1*). All Vietnam patients were admitted to one of two tertiary care referral hospitals in Ho Chi Minh City, Vietnam: Pham Ngoc Thach Hospital for Tuberculosis and Lung Disease (designated Hospital 1) and the Hospital for Tropical Diseases (designated Hospital 2) (*Heemskerk et al., 2016*). All patients were treated with adjunctive dexamethasone for the first 6–8 weeks with the regimen adjusted to disease severity on presentation (*Thwaites et al., 2004*).

Patient cohorts were compared overall as well as stratified into disease severity groups based on the TBM grade and by *LTA4H* genotypes, into the TT group (previously linked to response to steroids) and the CC, CT, and combined CC and CT genotypes (non-TT group). The Bayesian analysis methods to compare survival probabilities are detailed in Appendix 2. We limited analysis to the first 9 months of the one-year observation period in Indonesia to be compatible with the 9 month

observation period in Vietnam. For the data in Tables 1 and 2, Bayesian posterior probabilities for the comparisons of genotype and GCS frequencies, for each bin separately and for the mean of the distribution, were based on a flat Dirichlet prior on the bin probabilities.

Comparisons of age and time to median mortality were done using comparisons of arithmetic means of distributions, allowing Bayesian model choice between the following distribution families: Gaussian, log-Gaussian, Student, log-Student, Gamma, inverse-Gamma, Gamma-power. For age, log-Gaussian had overall the highest posterior probability; for time to median mortality, inverse-Gamma was preferred.

## Acknowledgements

We thank R Troll for evaluating the two published cohort studies, realizing that Bayesian analysis could provide answers and initiating the collaboration with the Bayesian statistician RS.

## Additional information

### Funding

| Funder | Grant reference number | Author |
|---|---|---|
| National Institutes of Health | NIAID ULTIMATE project 1R01AI145781-01 | Arjan van Laarhoven Reinout van Crevel |
| Wellcome Trust | Wellcome International Intermediate Fellowship 206724/Z/17/Z | Nguyen Thuy Thuong Thuong |
| Wellcome Trust | 110179/Z/15/Z | Guy E Thwaites |
| Wellcome Trust | Principal Research Fellowship 103950/Z/14 | Lalita Ramakrishnan |
| National Institutes of Health | R37 AI054503 | Lalita Ramakrishnan |

The funders had no role in study design, data collection and interpretation, or the decision to submit the work for publication.

### Author contributions

Laura Whitworth, Data curation, Formal analysis, Writing - original draft, Writing - review and editing; Jacob Coxon, Software, Writing - review and editing; Arjan van Laarhoven, Nguyen Thuy Thuong Thuong, Sofiati Dian, Bachti Alisjahbana, Data curation, Writing - review and editing; Ahmad Rizal Ganiem, Reinout van Crevel, Writing - review and editing; Guy E Thwaites, Mark Troll, Paul H Edelstein, Methodology, Writing - review and editing; Roger Sewell, Data curation, Software, Formal analysis, Supervision, Methodology, Writing - original draft, Project administration, Writing - review and editing; Lalita Ramakrishnan, Conceptualization, Resources, Formal analysis, Supervision, Funding acquisition, Visualization, Writing - original draft, Project administration, Writing - review and editing

### Author ORCIDs

Laura Whitworth (ID) https://orcid.org/0000-0002-8232-4601
Arjan van Laarhoven (ID) http://orcid.org/0000-0002-6607-4075
Guy E Thwaites (ID) http://orcid.org/0000-0002-2858-2087
Paul H Edelstein (ID) https://orcid.org/0000-0002-4069-5279
Roger Sewell (ID) https://orcid.org/0000-0003-4267-7055
Lalita Ramakrishnan (ID) https://orcid.org/0000-0003-0692-5533

### Decision letter and Author response
Decision letter https://doi.org/10.7554/eLife.61722.sa1
Author response https://doi.org/10.7554/eLife.61722.sa2

## Additional files

### Supplementary files

- Source code 1. Inference software.
- Source data 1. Anonymised patient dataset.
- Supplementary file 1. Characteristics of included and excluded Indonesia patients.
- Transparent reporting form

### Data availability

Excel spreadsheets with all patient data are included.

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

## Appendix 1

### Comparison of Bayesian and frequentist methods

### 1. Introduction

We give here a brief comparison of Bayesian methods with the more commonly encountered frequentist methods familiar to all scientists.

For context we consider a situation where we have collected some data $x$ (potentially a vector of many variables) involving two sets of patients $A$ and $B$, and want to determine whether an underlying parameter $a$ of set $A$ is greater or smaller than the corresponding parameter $b$ of set $B$; these form our two hypotheses $H_0$ (that $a \leq b$) and $H_1$ (that $a > b$).

We describe first the primary characteristics of each method, and then consider the consequences for the further properties they display.

### 2. The Bayesian method

The Bayesian method answers the question:

> What is the probability that $a > b$ given the data ?

giving answers of the form

> The probability given the data that $a > b$ is 0.96.

or (with the same meaning)

> The posterior probability that $a > b$ is 0.96.

In order to answer this question, the Bayesian method requires as input (a) the data $x$ and (b) probability distributions on each of the unknown parameters which describe what was known about their values before collecting the data. The latter are known as the "prior(s)". This is in keeping with reality — what we know after collecting some data depends not only on the data but also on what we knew before collecting the data. The answer is then calculated using Bayes' theorem.

Note that it is not possible to say 'We know nothing about the unknown parameters', and indeed this is never the case (e.g. if the unknown parameter is probability of survival at one year, then we know it is between 0 and 1, and unlikely to be either 0 or 1 exactly). However, one can almost always (and in particular in the present context) give distributions which leave wide open the range of values the parameters could take.

As a result, a Bayesian analysis, in addition to reporting the results, also reports the priors used, and what difference changing the priors makes to the results — in this case see sections 2.3 and 5 of Appendix 2 for these points. If priors are chosen to permit a broad range of parameter values (as in the present paper), it is usually the case that changing them to any other such prior has little effect on the results, so long as there is a reasonable quantity of data.

### 3. The frequentist method

The frequentist method aims to (a) avoid having to consider what was known before collecting the data, and (b) to control the probability of deciding $H_1$ is true if in fact $H_0$ is true (the 'type I error rate'). To achieve this aim it answers a more complicated question; we give a version of it specific to a particular level of frequentist 'confidence':

> Did the data $x$ fall into a critical region $C$, which we chose before collecting any of the data, such that, for any set of values $h$ of the unknown parameters for which $a \leq b$, the probability given $h$ of $x$ falling into $C$ is at most 0.04 ?

If the answer is Yes, this fact is usually encoded in a statement such as

> We are 96% (frequentist-)confident that $a > b$ by the one-sided Student $t$-test.

or just

> $a > b$ ($p = 0.04$, one-sided Student $t$-test).

Here the $p$ value is usually one minus the frequentist confidence, and 'one-sided Student $t$-test' specifies a particular standard set of nested critical regions (one for each level of confidence). For all but the simplest of frequentist tests, the user only rarely attempts to visualise or imagine the nested family of critical regions, which are often/usually multidimensional, as would be true in the present case were we to use frequentist methods.

In doing this the frequentist method treats $H_0$ differently from $H_1$, and thereby replaces the Bayesian's prior with an asymmetry on the hypotheses that doesn't correspond to any prior.

## 4. Consequent differences in behaviour

The frequentist's avoidance of using a prior and emphasis on controlling the type I error rate together incur a significant cost.

First, it is an absolute requirement of the frequentist method that the critical region be specified *before* the data is collected. However, this requirement is very often ignored, and this can lead to bias in a *post-hoc* analysis, for example if the investigator first tries a Mann-Whitney $U$-test, finds the result not significant, so tries a Student $t$-test, finds it significant, and so only reports the latter. Where the unknowns are more than 1-dimensional, as in the present survival analyses, this ambiguity gets exponentially worse with the number of dimensions.

The Bayesian method has no such restriction: a data set may be analysed multiple times *post-hoc* with different priors either to investigate the effect of changing the prior or because different workers have different prior beliefs.

However, both methods can potentially be biased by omitting to report findings which are not to the investigator's liking. In this paper, the only important omission is that analysis was also done of the HIV-positive patients in Vietnam, and this was not reported in this paper. This was because, although it tended to confirm similar findings to those from the HIV-negative cohort, there were some points that were less clearcut and harder to interpret, and its publication therefore awaits completion of a clinical trial in which the benefit of dexamethasone is being examined for HIV-positive patients of all three genotypes by randomizing them to get dexamethasone or not (*Donovan et al., 2018*).

Second, repeated analyses at different stages of data collection and/or analyses of multiple subsets of the patients *post-hoc* increase the type I error rate, and therefore incur penalties in the frequentist method, making it more difficult for any individual comparison to reach significance. In contrast the Bayesian method controls the posterior probability (which is unaffected by other independent analyses being done), and not the type I error rate. The effects of this difference are summarised in *Box 1* in the main paper.

Third, frequentist analysis fails to respect common-sense real-world laws of inference. For example, it is often the case that a frequentist analysis will find that it is e.g. 95%-confident that $a$ is non-zero, but 97.5%-confident that $a>0$, a stronger statement than that $a \neq 0$. In contrast Bayesian methods, being rigorously derived from probability theory, obey the laws that probability already obeys.

Fourth, frequentist analysis gives priority to controlling the type I error rate (i.e. the probability that it concludes $H_1$ holds if in fact $H_0$ holds), and almost never controls the probability that it concludes that $H_0$ holds if in fact $H_1$ holds. (Some frequentist analyses do report 'power calculations', which calculate the probability that it concludes that $H_0$ holds if in fact *some subset* of $H_1$ holds, e.g. the part of $H_1$ where $a>b+10$ rather than that just $a>b$.) In contrast Bayesian methods treat $H_0$ and $H_1$ identically.

## Appendix 2

### Supplementary methods

1. Overview of methods for survival analysis

We adopt the Bayesian paradigm. (For those who haven't encountered Bayesian methods before, a comparison with frequentist hypothesis testing is given in Appendix 1.) Accordingly we define below a probabilistic generative model for patient lifetime $x$ given model parameters $\theta$ and a suitable prior distribution on $\theta$. We assume that, when observed, such lifetime data may be censored at known times $t$ that are independent of the underlying lifetimes and the model parameters, so that for each patient we observe either a time of death $x$ or a time of censoring $t$ at which the patient was alive.

We then collect a data set of values $\hat{\mathbf{x}}$ of $x$ or $t$ for a set of patients, and apply Bayes' theorem to deduce the posterior distribution $P(\theta|\hat{\mathbf{x}})$ of the model parameters.

Since this distribution is hard to visualise, we draw samples of $\theta$ from it using Markov chain Monte Carlo (MCMC) methods described below. For each such sample of $\theta$ we calculate the distribution of lifetime $x$ given $\theta$ and hence the survival probability $q(t|\theta)$ and hazard rate $h(t|\theta)$ functions against patient time since admission, using the following relationships:

$$q(t|\theta) = P(x \geq t|\theta) = 1 - \int_0^t P(x|\theta)\,dx$$

$$h(t|\theta) = -\frac{d}{dt}\log(q(t|\theta)).$$

An example resulting set of survival probability curves is shown in *Appendix 2—figure 1*, with the corresponding hazard rate curves being shown in *Appendix 2—figure 2*.

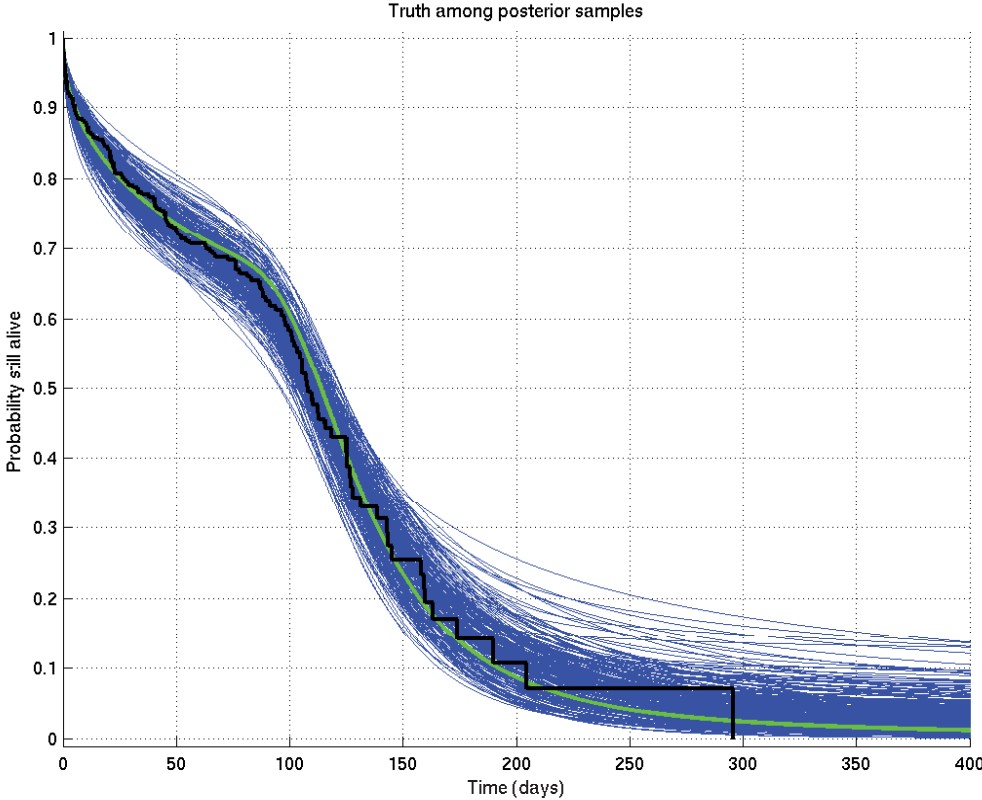

**Appendix 2—figure 1.** Output of test run using synthetic data for which the right answer is known. The true survival probability curve is shown in green, with the Kaplan-Meier plot for the generated data in black. In blue are shown many samples from the posterior distribution on the survival

*Appendix 2—figure 1 continued on next page*

*Appendix 2—figure 1 continued*

probability curve, calculated from $P(\theta|\hat{\mathbf{x}})$, which indicate the uncertainty in the inferred distribution. The synthetic dataset comprised 300 hypothetical patients of whom the time of death of 153 was censored.

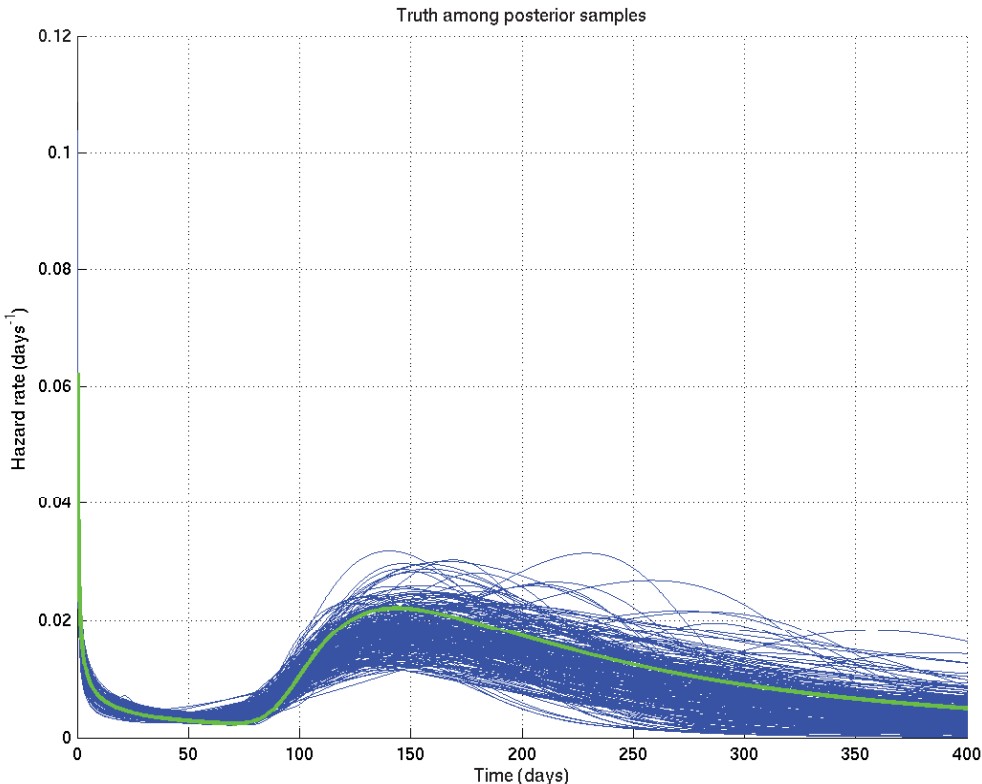

**Appendix 2—figure 2.** Output of test run using synthetic data for which the right answer is known. The true hazard rate curve is shown in green. In blue are shown many samples from the posterior distribution on the hazard rate curve, calculated from $P(\theta|\hat{\mathbf{x}})$, which indicate the uncertainty in the inferred distribution. The synthetic dataset comprised 300 hypothetical patients of whom the time of death of 153 was censored.

Now, given a set of such curves, we can calculate the mean posterior survival probability (resp. hazard rate) at each time point and plot the posterior mean survival probability (resp. hazard rate) against time. Similarly we can find the 2.5% and 97.5% centiles. Examples of the posterior mean and centile curves for survival probability corresponding to *Appendix 2—figure 1* are shown in *Appendix 2—figure 3*.

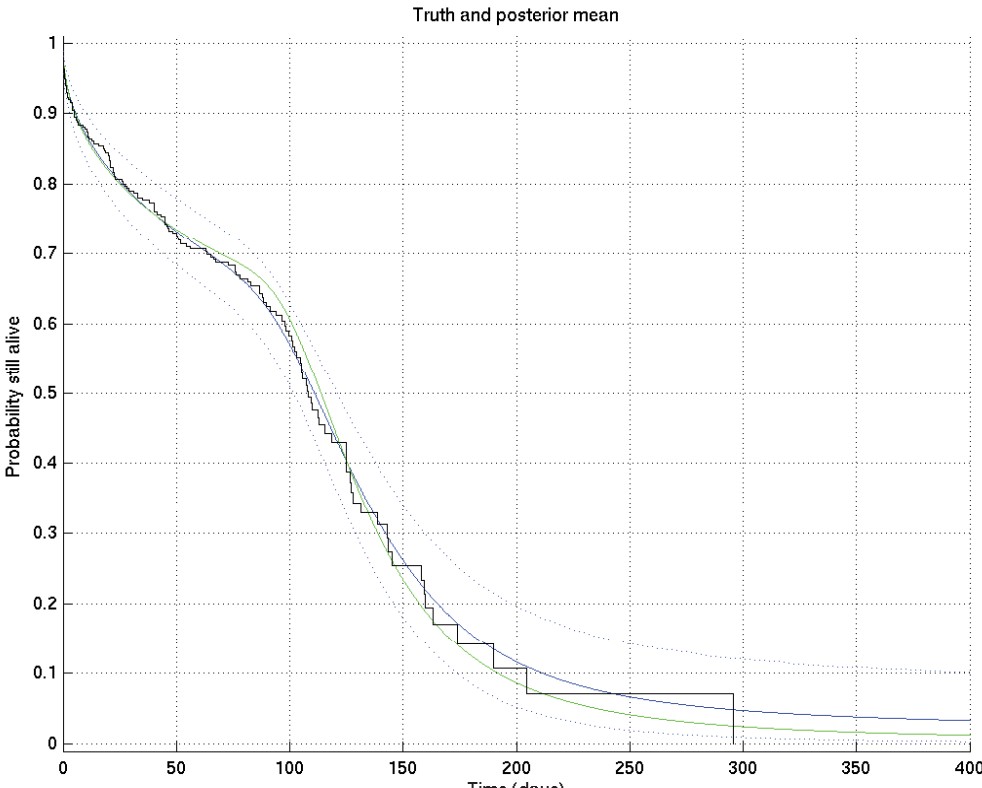

**Appendix 2—figure 3.** Output of test run using synthetic data for which the right answer is known. The true survival probability curve is shown in green, with the Kaplan-Meier plot corresponding to the generated data in black. In blue is the posterior mean survival probability against time, calculated from $P(\theta|\hat{\mathbf{x}})$, and in dotted lines the 2.5% and 97.5% centiles, which indicate the uncertainty in the inferred distribution. The synthetic dataset comprised 300 hypothetical patients of whom the time of death of 153 was censored.

Further, given two such sets of survival probability or hazard rate curves (such as the survival probability curves shown in *Appendix 2—figure 4* and zoomed in in *Appendix 2—figure 5*), we can also calculate at each time point the probability that a value taken at that time from a random curve from set 1 is greater than a similar value taken from a random curve from set 2. As we can see from these two figures, this probability is higher at 10 days than at 1 day or 100 days. These probabilities, at each time point, can then be plotted giving in this example *Appendix 2—figure 6*, showing the probability that underlying survival in group 1 is greater than that in group 2 at each time point.

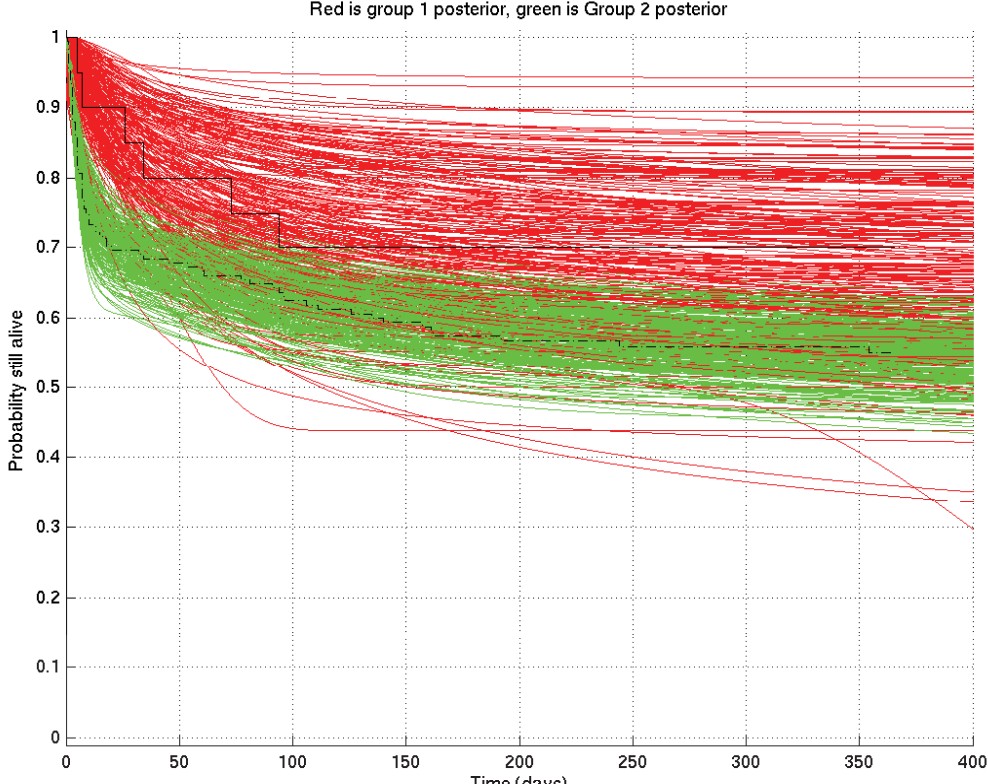

**Appendix 2—figure 4.** Comparison of two runs generated independently from two subsets of patients, subset 1 (red) consisting of 20 patients and subset 2 (green) consisting of 182 patients. The Kaplan-Meier plot for subset 1 is in solid black and that for subset 2 in dot-dashed black. Since there were many more patients in subset 2 than in subset 1, we expect greater variance in the inferred survival probabilities for subset 1 than for subset 2.

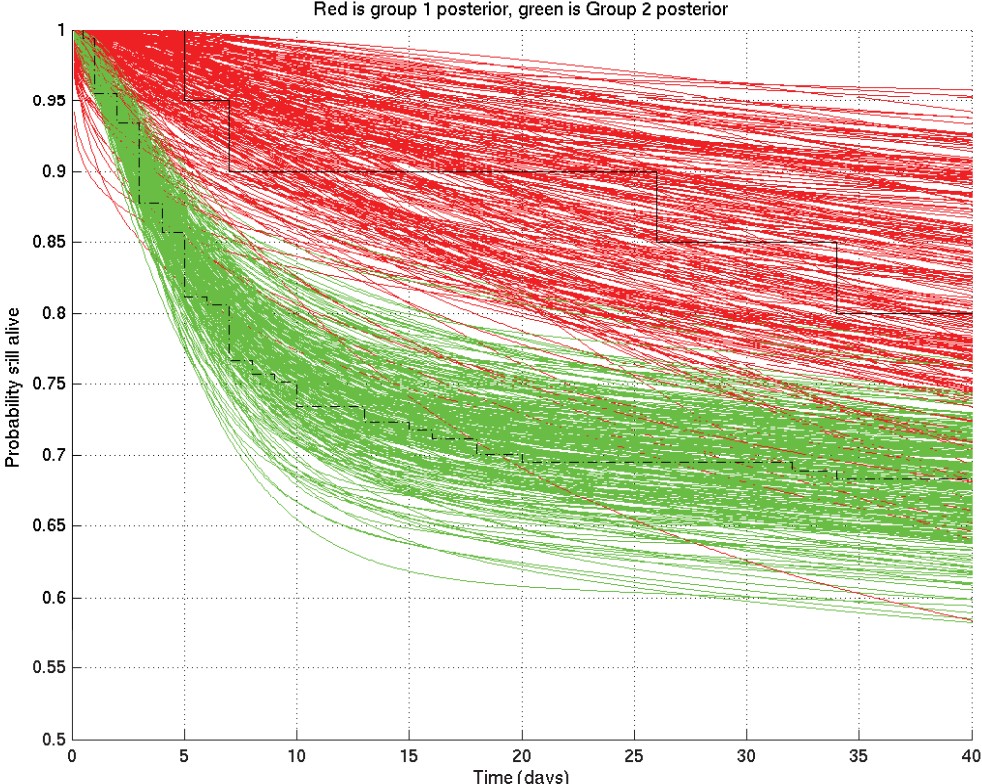

**Appendix 2—figure 5.** As for *Appendix 2—figure 4* but zoomed in to the top left hand corner, showing greater separation and less overlap of the red and green curves at 10 days than at 1 day or 100 days.

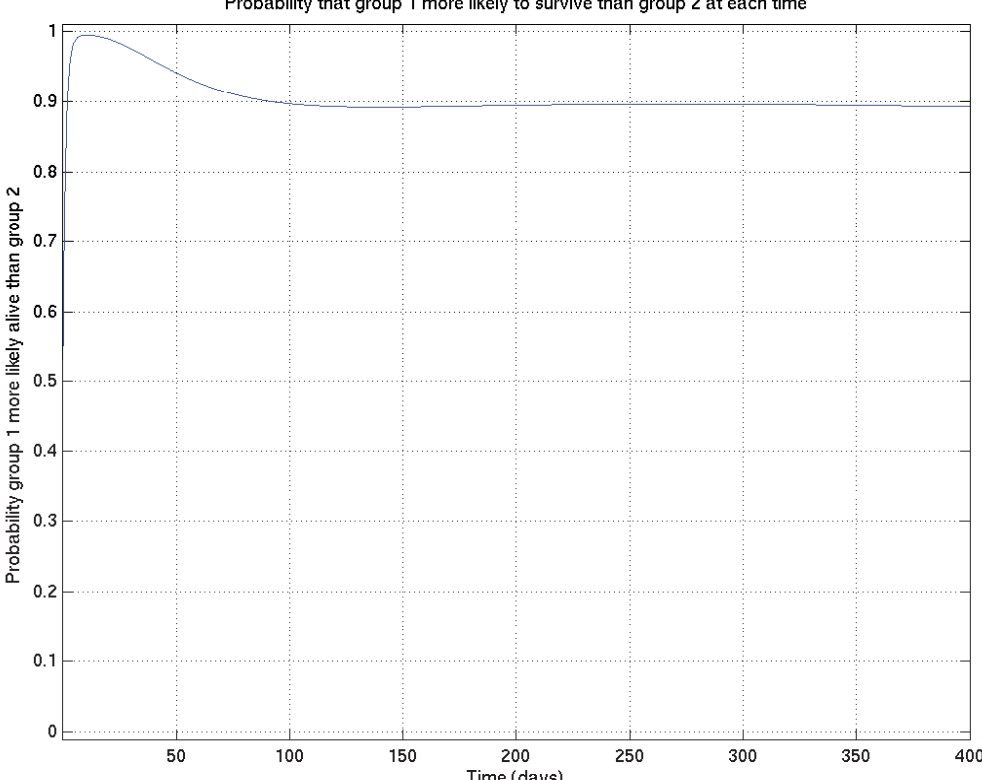

**Appendix 2—figure 6.** Corresponding to *Appendix 2—figures 4* and *5* this shows the probability that survival for subset 1 is greater than that for subset 2 at each time point.

In order to carry out this process, we need to specifically define the lifetime distribution model $P(x|\theta)$, and we now turn to this.

## 2. Lifetime model

We suppose that there exist an unknown number $j$ of different mechanisms causing death, and that each such mode of death has a different lifetime distribution.

### 2.1 Combination of different modes of death

Let $x$ denote a lifetime, that is, the time until a patient dies. Let $j \in \{1, 2, ..., J\}$ denote a particular mechanism (or mode) of death. Let $x_j$ denote the time at which mode $j$ would kill the patient; we set $x_j = \infty$ to denote the possibility that that mode would never have killed the patient.

Then the patient's time of death is given by

$$x = \min_{1 \leq j \leq J} x_j.$$

In particular $x = \infty$ denotes the situation that the patient never dies (unlikely as this is).

### 2.2 Model of a single mode of death

We now drop the subscripts $j$, but assume that this subsection will be repeated $J$ times with the subscript $j$s added to every random variable, with each of the repetitions being independent as far as the model is concerned before being conditioned on observed data. When later we want to refer to the complete set of $J$ values of e.g. $p$, we will use bold face, e.g. $\mathbf{p} = (p_1, p_2, ..., p_J)$.

Thus we will set $P(x|p, k, m, r)$, i.e. $P(x_j|p_j, k_j, m_j, r_j)$, to be such that with probability $p$, $x^k$ is Gamma distributed with parameters $m' = m$ and $r' = mr^k$, and otherwise $x = \infty$. Thus we have

$$P(x|p,k,m,r) = \begin{cases} p|k|\frac{(mr^k)^m}{\Gamma(m)}x^{mk-1}e^{-m(rx)^k} & (0<x<\infty) \\ 1-p & (x=\infty) \\ 0 & (x \leq 0). \end{cases}$$

Here $p \in [0,1]$, $k \in \mathbb{R} \setminus \{0\}$, $m,r > 0$.

Note that we have here a distribution which has both a discrete and a continuous part, so that $P(x|...)$ is used as notation both for a probability and for a probability density: in other words, we have a continuous distribution for finite positive $x$, given by a density function, whose interpretation is that its integral from $x_1$ to $x_2$ is the probability that $x_1 < x < x_2$; but as we have a non-zero probability $1-p$ that $x = \infty$, the integral from 0 (inclusive) to $\infty$ (exclusive) of the density given by the first line of the above formula for $P(x|p,k,m,r)$ must be $p$. On the other hand we have a discrete distribution for $x = \infty$, and $1-p$ is a probability, not a density.

By way of very approximate intuition: $p$ is the probability that a particular mode of death would kill the patient at a finite time; $r$ is the reciprocal of the overall timescale to deaths of those patients who die; $m$ governs how variable those times of death are – the smaller $m$ is, the more variable are the times of death; and the sign of $k$ plays a part in determining whether the hazard rate for this mode of death is increasing or decreasing, while the magnitude of $k$ governs how abruptly the spread of death time is cut off in the less spread out direction. Specifically, $k = 0$ makes no sense, as then we would have $x^k = 1$ for all $x$, and an invalid distribution would result (so it should not be a surprise that the prior on $k$ is bimodal with zero density at zero).

## 2.3 Priors on the parameters

We specify the priors on the parameters in two stages. First, we specify their general form, and second we choose specific values for the hyperparameters that then specify a unique prior.

### 2.3.1 General form of the priors

The total number $j$ of modes is itself to be considered a random variable, on which we put the prior

$$P(J|\alpha_{\mathrm{J}}) = (1-\alpha_{\mathrm{J}})\alpha_{\mathrm{J}}^{J-1}$$

for $J \in \mathbb{N}^* = \{1,2,...\}$ and for some fixed $\alpha_{\mathrm{J}} \in [0,1)$.

The prior for the parameters $p,m,r,k$ of each mode of death are taken to be independent, and as follows.

We take the prior on $p$ to be Beta, with positive real parameters $\alpha_{\mathrm{p}}, \beta_{\mathrm{p}} > 0$, so that

$$P(p|\alpha_{\mathrm{p}},\beta_{\mathrm{p}}) = \frac{\Gamma(\alpha_{\mathrm{p}}+\beta_{\mathrm{p}})}{\Gamma(\alpha_{\mathrm{p}})\Gamma(\beta_{\mathrm{p}})}p^{\alpha_{\mathrm{p}}-1}(1-p)^{\beta_{\mathrm{p}}-1}.$$

We take the prior on $r$ to be Gamma, with parameters $m_{\mathrm{r}}, r_{\mathrm{r}} > 0$, so that

$$P(r|m_{\mathrm{r}},r_{\mathrm{r}}) = \frac{r_{\mathrm{r}}^{m_{\mathrm{r}}}}{\Gamma(m_{\mathrm{r}})}r^{m_{\mathrm{r}}-1}e^{-r_{\mathrm{r}}r}.$$

We take the prior on each of the parameters $k$ and $m$ to be the conjugate prior on each with respect to this parameterisation. Thus for positive real parameters $a_{\mathrm{m}}, b_{\mathrm{m}}$ we have

$$P(m|a_{\mathrm{m}},b_{\mathrm{m}}) \propto \frac{m^{b_{\mathrm{m}}m}e^{-(a_{\mathrm{m}}+b_{\mathrm{m}})m}}{\Gamma(m)^{b_{\mathrm{m}}}}.$$

Similarly for parameters $N_{\mathrm{k}} \in \mathbb{N}$, $a_{\mathrm{k}} \in \mathbb{R}_+$ and $\mathbf{b}_{\mathrm{k}} = (b_1,b_2,...,b_{N_{\mathrm{k}}})$, $\mathbf{c}_{\mathrm{k}} = (c_1,c_2,...,c_{N_{\mathrm{k}}}) \in \mathbb{R}_+^{N_{\mathrm{k}}}$ we have

$$P(k|a_{\mathrm{k}},\mathbf{b}_{\mathrm{k}},\mathbf{c}_{\mathrm{k}}) \propto |k|^{a_{\mathrm{k}}}\prod_{n=1}^{N_{\mathrm{k}}}b_n^{kc_n}e^{-c_nb_n^k}.$$

## 2.3.2 Specific values of the hyperparameters and the resulting priors

Specific values were chosen for the hyperparameters by varying them and showing those users overseeing the analysis (namely LW, MT, PE, LR, of whom PE and LR are infectious diseases clinicians) the resulting distributions of samples of survival curves and hazard rates, then letting them choose the most appropriate prior given their background experience. The prior chosen was intentionally uninformative and very wide, while still being centred on the clinician-expected survival curves and hazard rates.

The specific parameter values chosen were as follows: $a_{\mathrm{J}} = 0.8$

$$a_{\mathrm{p}} = b_{\mathrm{p}} = 0.5$$

$$a_{\mathrm{m}} = b_{\mathrm{m}} = 1$$

$$m_{\mathrm{r}} = 0.5$$

$$r_{\mathrm{r}} = 30 \, \mathrm{days}$$

$$N_{\mathrm{k}} = 2$$

$$a_{\mathrm{k}} = 1$$

$$\mathbf{b}_{\mathrm{k}} = (0.2, 0.2)$$

$$\mathbf{c}_{\mathrm{k}} = (0.5, 0.5).$$

These result in the following depicted distributions for $J$, $p_j$, $m_j$, $r_j$, $k_j$, and hence for the depicted samples from the distributions for survival probability and hazard rate against time as well as the mean and 2.5% and 97.5% centiles for the last two: see *Appendix 2—figures 7 to 15*.

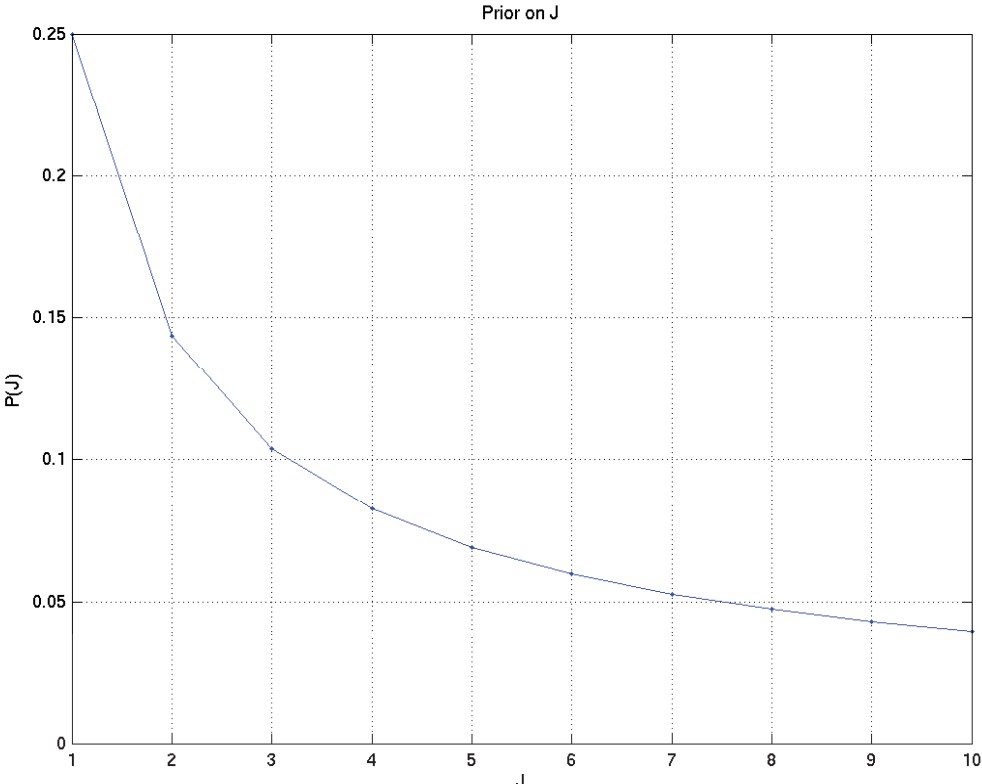

**Appendix 2—figure 7.** Prior on $J$, the number of different modes of death.

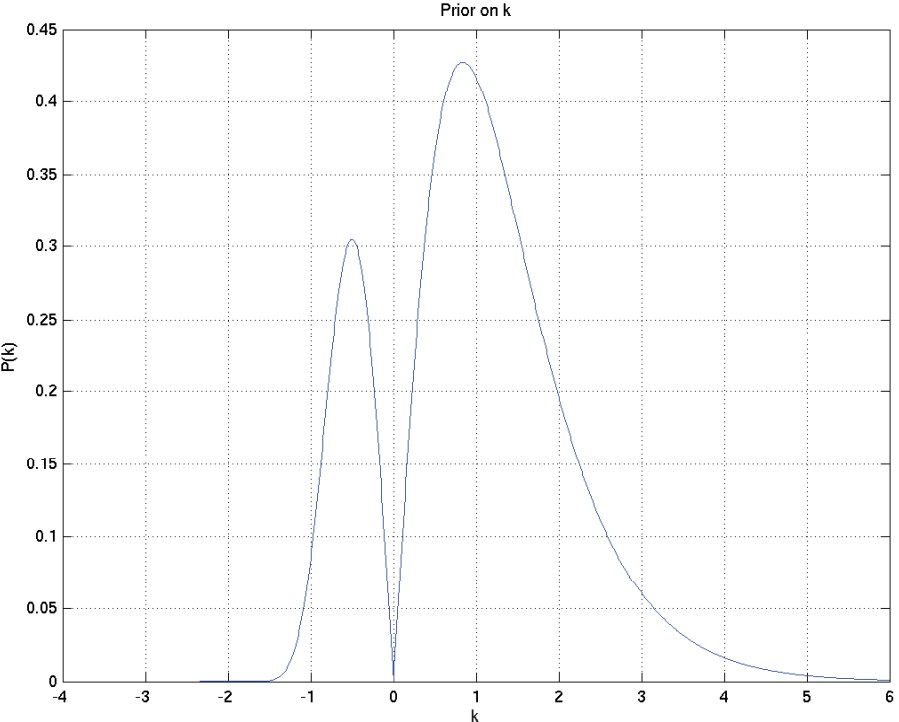

**Appendix 2—figure 8.** Prior on $k$.

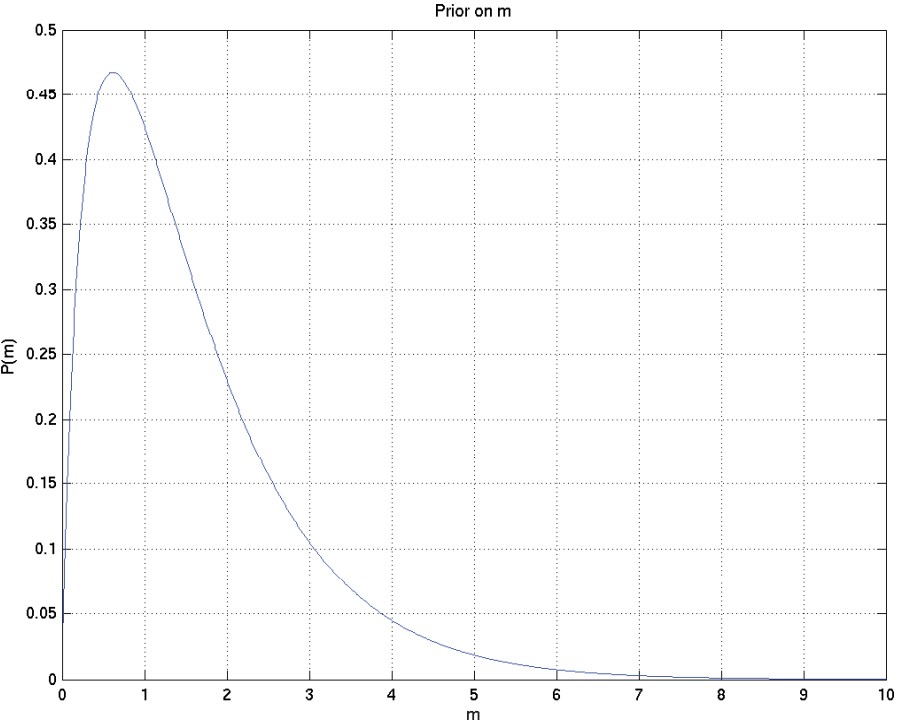

**Appendix 2—figure 9.** Prior on $m$.

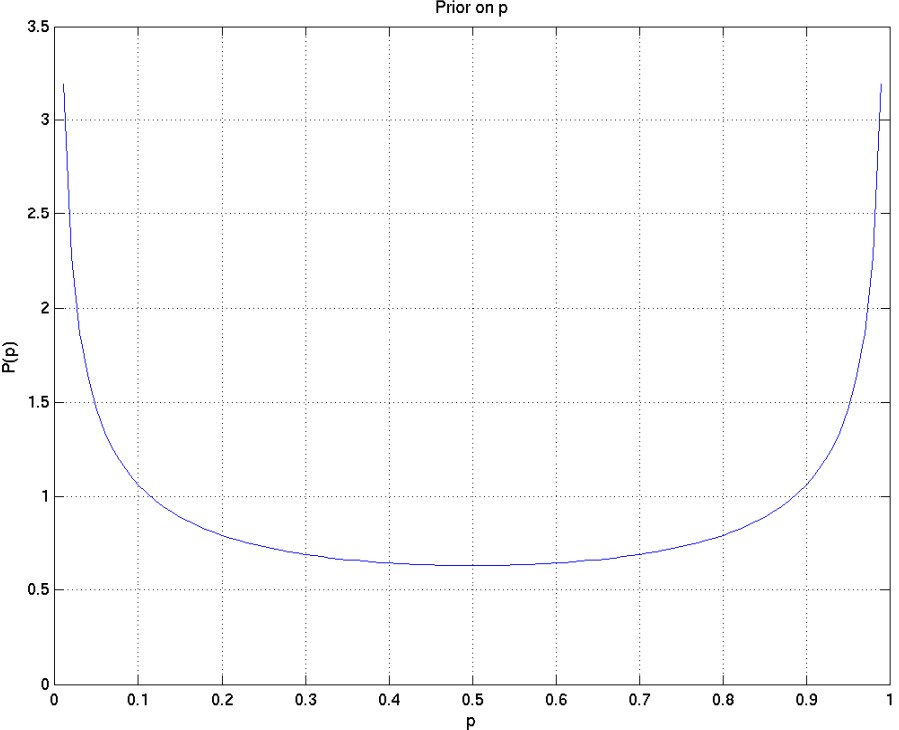

**Appendix 2—figure 10.** Prior on $p$.

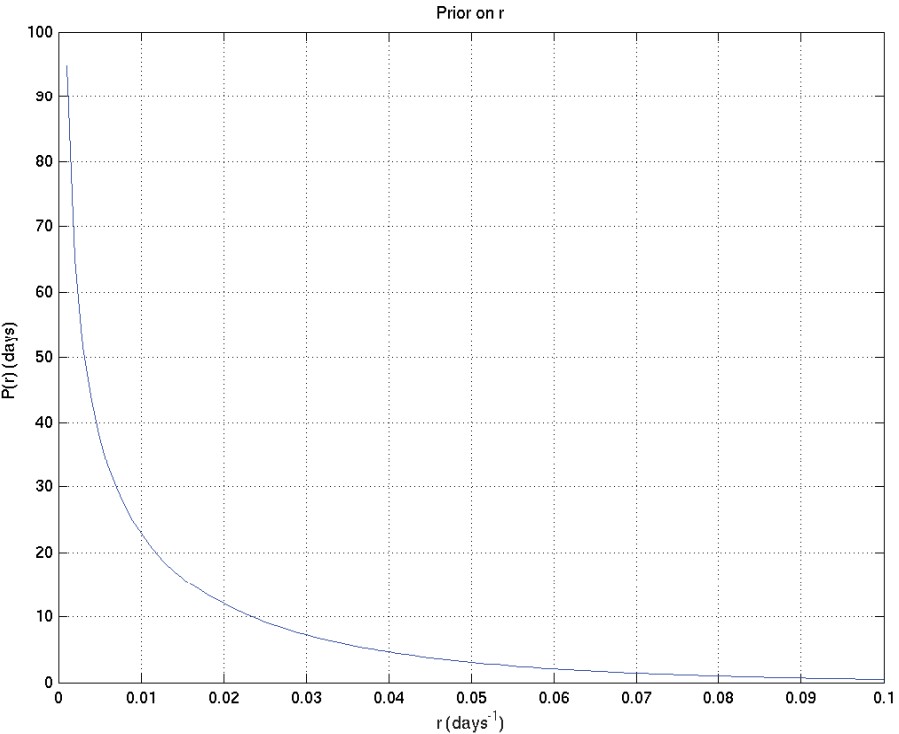

**Appendix 2—figure 11.** Prior on $r$.

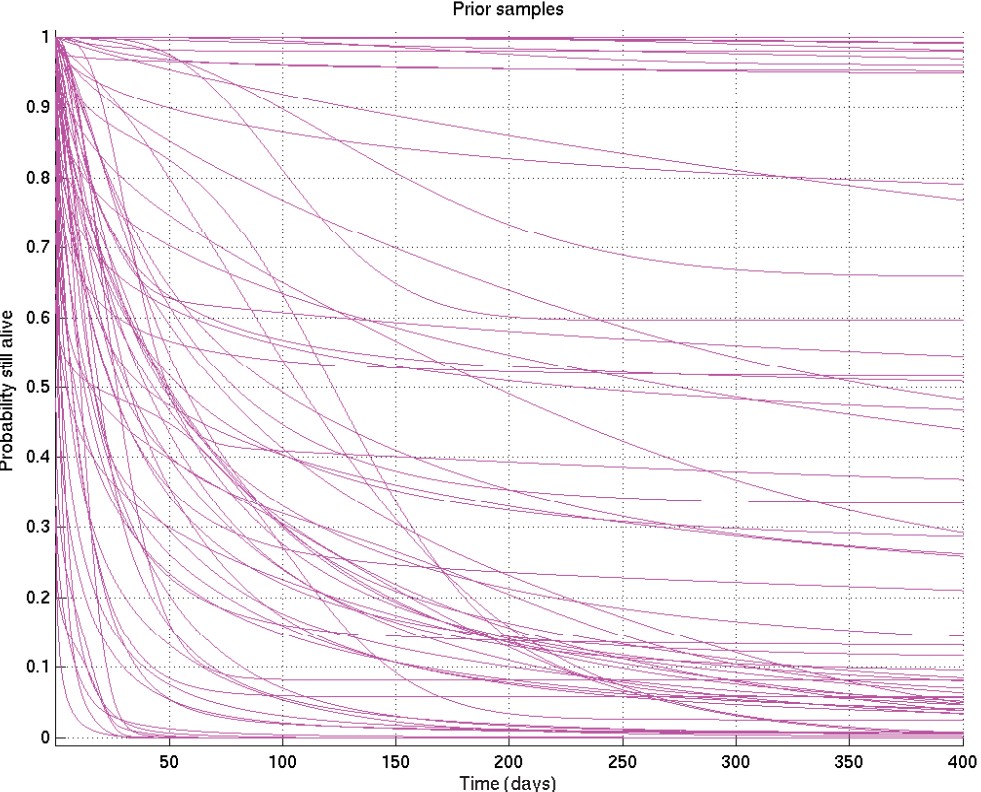

**Appendix 2—figure 12.** Samples from resulting prior on survival probability against time.

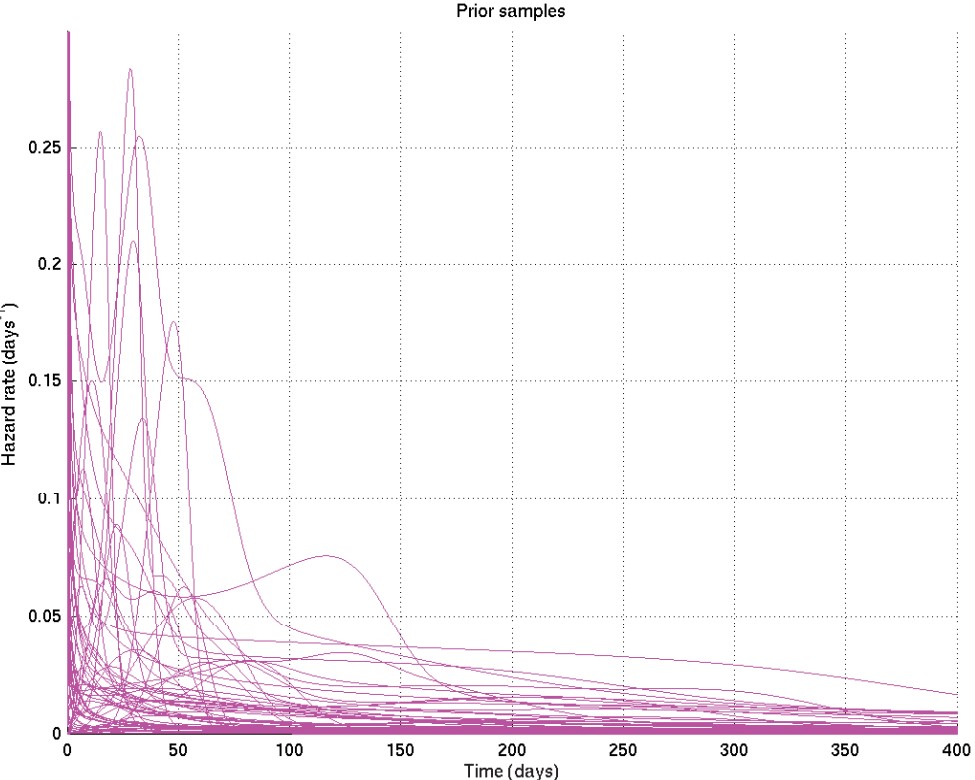

**Appendix 2—figure 13.** Samples from resulting prior on hazard rate against time.

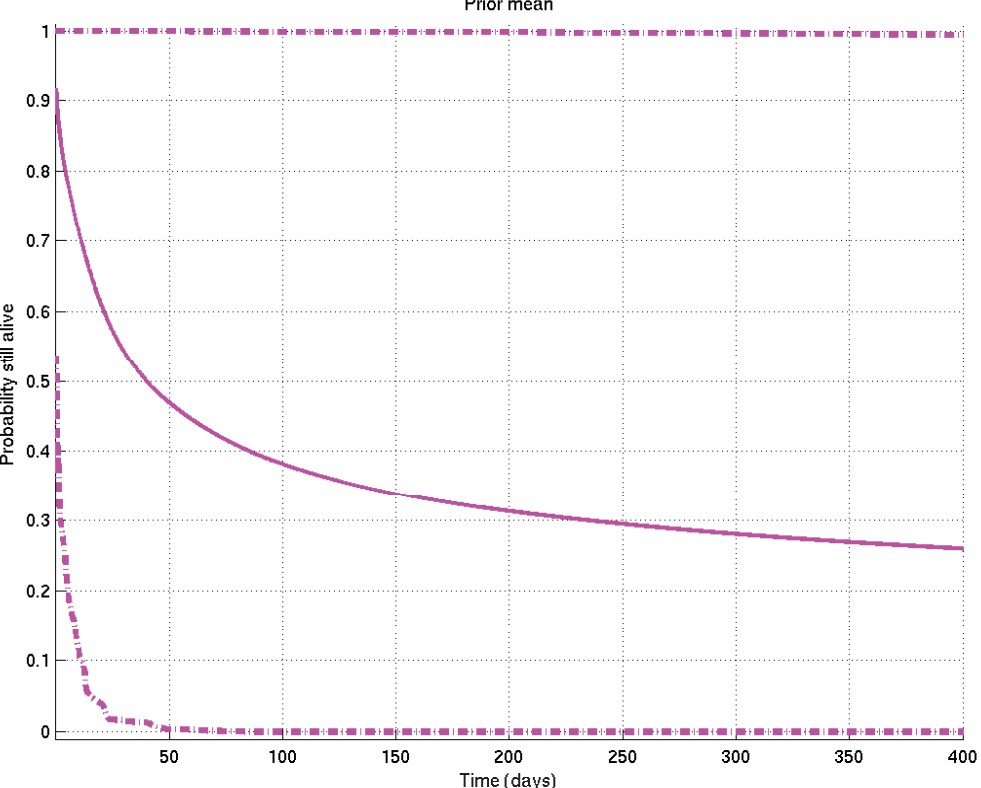

**Appendix 2—figure 14.** Mean and 2.5% and 97.5% centiles of prior on survival probability against time.

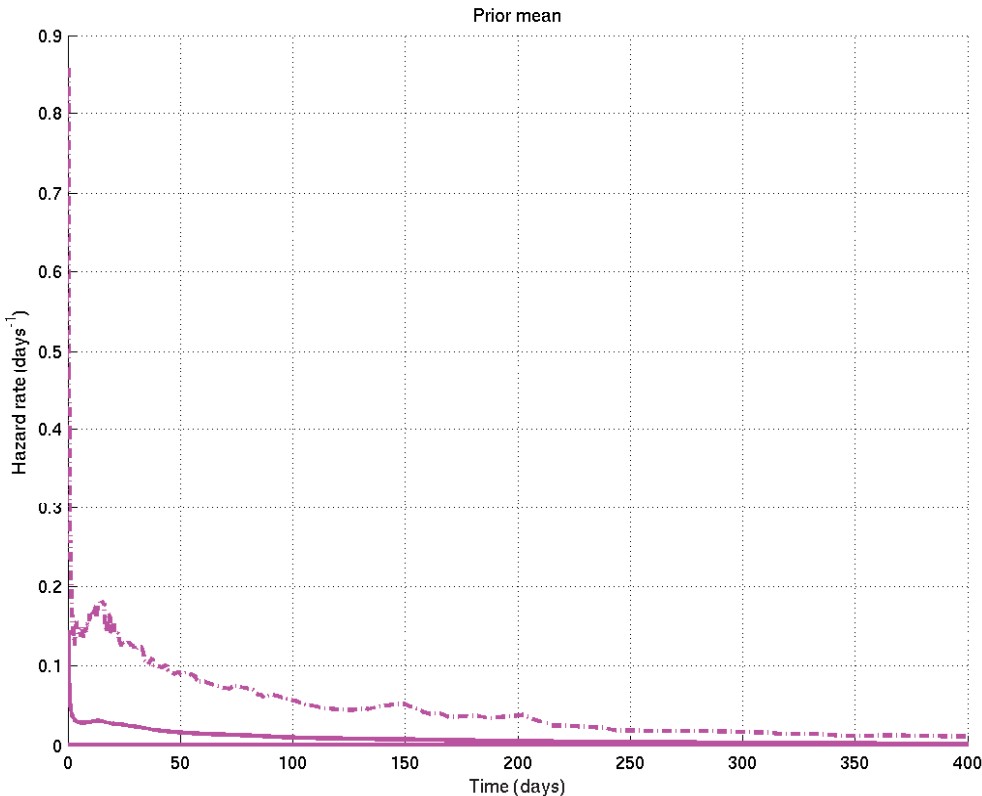

**Appendix 2—figure 15.** Mean and 2.5% and 97.5% centiles of prior on hazard rate against time.

## 2.4 Summary

Thus the parameters $\theta$ consist of $\theta = (J, \mathbf{p}, \mathbf{m}, \mathbf{r}, \mathbf{k})$, and the prior

$$P(\theta) = P(J) \prod_{j=1}^{J} P(p_j) P(m_j) P(r_j) (k_j),$$

a combination of densities of different dimensionalities. This can be regarded a mixture of models, one for each possible value of $J$, weighted by the prior on $J$: the model for $J = 1$ has only one possible mode of death, while that for $J = 2$ has two possible modes of death, which are independent, and so on for higher values of $J$.

In particular, when comparing two subsets A and B of patients, we use the same prior to infer the posterior distribution of $\theta$ given $\hat{\mathbf{x}}$ for each subset of the patients independently. Thus a priori the probability at each time $t$ that $q_A(t) > q_B(t)$ is $\frac{1}{2}$, the probability that $q_A(t) < q_B(t)$ is $\frac{1}{2}$, and the probability that $q_A(t) = q_B(t)$ is zero (though it has non-zero probability density). In other words we are asking how sure we are that $q_A(t) > q_B(t)$ given the data, reckoning the alternative to be that $q_A(t) < q_B(t)$, and taking for granted that the probability that the two are *exactly* the same is zero.

## 2.5 Rationale

The basic Gamma model is a frequently used model of a failure mode that goes through several stages of failing, each with an exponentially distributed lifetime, before final failure occurs. The additional effect of the parameter $k$ is to incorporate Weibull-type failure time models, allowing for both increasing and decreasing hazard rates. The parameter $p$ allows for the possibility that some modes of dying may not be relevant for all patients, while the combination of the $J$ submodels allows for a number of different types of mechanisms of death to be relevant.

In particular, we specifically used independent priors on the parameters of the subsets being compared because:

- When offered the choice the users (namely LW, MT, PE, LR, GT of whom PE, LR, and GT are infectious diseases clinicians with GT having extensive experience with TB meningitis patients) indicated unanimously that this correctly represented their prior beliefs;
- Because we restrict our use of posterior probabilities to those of the form $P(U>V|\text{data})$ rather than $P(U>V+\alpha|\text{data})$ for some $\alpha>0$, and because the additional effect of any non-independent factors in the prior would be symmetric either side of the difference between the two subsets being zero, the effect of any non-independent factors would be expected to be very small;
- If they did not believe the priors on the relevant disjoint subsets were independent, the clinicians involved would find it very hard to specify exactly how similar each pair of subsets being compared should be expected to be.

## 3 MCMC methodology

We introduce additional variables $j_i$ for each patient $i$ which indicate whether the time of death was censored (value 0) or was caused by a particular mode $j$ of death (value $j \neq 0$ unknown). We also introduce variables $x_{i,j}$ of unknown values giving for each patient the time of death that would have resulted from mode $j$ if no other modes had killed the patient first. These variables take the specific value $x_{i,j} = \infty$ if mode $j$ would in fact not have killed patient $i$ at any finite time.

We initialise the parameters $J, \mathbf{p}, \mathbf{m}, \mathbf{r}, \mathbf{k}$ from the prior and initialise the additional variables $\mathbf{j}$ and $\mathbf{x}$ randomly to any set compatible with those and the observed variables $\hat{\mathbf{x}}$. These variables then form $\theta_1 = (J, \mathbf{p}, \mathbf{m}, \mathbf{r}, \mathbf{k}, \mathbf{j}, \mathbf{x})$, the first of a sequence of samples $(\theta_n)_{n=1,...}$ to be drawn.

### 3.1 Sampling methods

A thorough review of all the following methods is available either in **Neal, 1993** or in **Dagpunar, 1988** except where otherwise indicated.

The key point is that if we resample each variable by a method that satisfies detailed balance, and given other weak conditions which are here fulfilled, Feller's theorem (**Neal, 1993**) then guarantees the sequence of samples $(\theta_n)$ will eventually converge to a sequence of samples from the desired distribution $P(\theta|\hat{\mathbf{x}})$. The samples in this sequence will not be independent of each other, though the conditional distribution of $\theta_{n_1}$ given $\theta_{n_0}$ will also converge to $P(\theta|\hat{\mathbf{x}})$ as $n_1 \to \infty$ with $n_0$ fixed, that is, to independence.

Sampling from the posterior was done by the MCMC technique of Gibbs sampling, that is, sampling from the following distributions palindromically:

1. $P(\mathbf{k}|\hat{\mathbf{x}}, \mathbf{x}, J, \mathbf{j}, \mathbf{m}, \mathbf{r}, \mathbf{p})$. This distribution has two parts ($k_j>0$ and $k_j<0$), each of which is log-concave. We therefore first resample the sign of each $k_j$ using the Metropolis-Hastings algorithm (**Neal, 1993**), then use adaptive rejection sampling (**Gilks, 2019**) to resample the magnitude of $k_j$ given its sign, then resample the sign again to maintain detailed balance.
2. $P(\mathbf{p}|\hat{\mathbf{x}}, \mathbf{x}, J, \mathbf{j}, \mathbf{m}, \mathbf{r}, \mathbf{k})$. In this case the conditional distribution is from the Beta family, and standard methods (**Dagpunar, 1988**) are available to sample from it.
3. $P(\mathbf{m}|\hat{\mathbf{x}}, \mathbf{x}, J, \mathbf{j}, \mathbf{r}, \mathbf{k}, \mathbf{p})$. This distribution is log concave, so we may use adaptive rejection (**Gilks, 2019**) sampling to sample from it.
4. $P(\mathbf{r}|\hat{\mathbf{x}}, \mathbf{x}, J, \mathbf{j}, \mathbf{m}, \mathbf{k}, \mathbf{p})$. For each $j$, this distribution is in general a product of a Gamma distribution on $r_j$ and a much narrower Gamma distribution on $r_j^{k_j}$. We therefore sample from the Gamma relevant to the latter (**Dagpunar, 1988**), using this as a proposal distribution for the Metropolis-Hastings algorithm (**Neal, 1993**), resulting in the Hastings ratio coming from the Gamma on $r_j$.
5. $P(\mathbf{j}|\hat{\mathbf{x}}, J, \mathbf{m}, \mathbf{r}, \mathbf{k}, \mathbf{p})$ then $P(\mathbf{x}|\hat{\mathbf{x}}, J, \mathbf{j}, \mathbf{m}, \mathbf{r}, \mathbf{k}, \mathbf{p})$. The first of these is a discrete distribution which is trivial to sample from, and the second reduces to a truncated Gamma distribution. To sample from the latter we divide into two cases: if the shape parameter is $\geq 1$ the distribution is log-concave and we can use adaptive rejection sampling (**Gilks, 2019**) otherwise we use Metropolis-Hastings (**Neal, 1993**) with either an exponential or a Gamma proposal distribution, depending which is estimated to be likely to be quicker given the other parameters.
6. $P(J|\hat{\mathbf{x}}, \mathbf{x}, \mathbf{j}, \mathbf{m}, \mathbf{r}, \mathbf{k}, \mathbf{p})$ (where only values of $j$ unused in $\mathbf{j}$ are allowed to be removed) followed, if $J$ has increased, by sampling the new elements of $\mathbf{m}, \mathbf{r}, \mathbf{k}, \mathbf{p}$ from the prior distributions on

these variables. Resampling of $J$ uses a discrete conditional distribution, and is done using a proposal to either increase or decrease $J$ by 1, and applying the appropriate Hastings ratio (**Neal, 1993**) to reject the proposal in such a way as to achieve detailed balance.

10,000 samples were drawn from the posterior for each subset of the data considered (e.g. for Indonesian TT patients). The first 1500 samples were discarded and the remainder kept for analysis. To check that the software was correct we undertook two types of check:

1. The inference code was reviewed by somebody (RFS) different from its author (JC) looking for bugs, and those found were removed after RFS and JC had conferred to reach agreement on them.
2. Multiple sets of synthetic data were generated (for which the true values of $\mathbf{x}, J, \mathbf{j}, \mathbf{m}, \mathbf{r}, \mathbf{k}, \mathbf{p}$ were therefore known) and the posterior distributions were compared with the true values (as for example in **Appendix 2—figure 1** above).

## 3.2 Convergence checks

In addition we checked for convergence of the Markov chains by starting them from different random initial values of $\mathbf{x}, J, \mathbf{j}, \mathbf{m}, \mathbf{r}, \mathbf{k}, \mathbf{p}$, then the two corresponding sets of output samples were compared as if they were made from two different sets of patients, giving plots analogous to **Appendix 2—figures 4** and **6** above, such as those shown in **Figures 16** and **17** below, which show that the two distributions are sufficiently close as to be in practice indistinguishable.

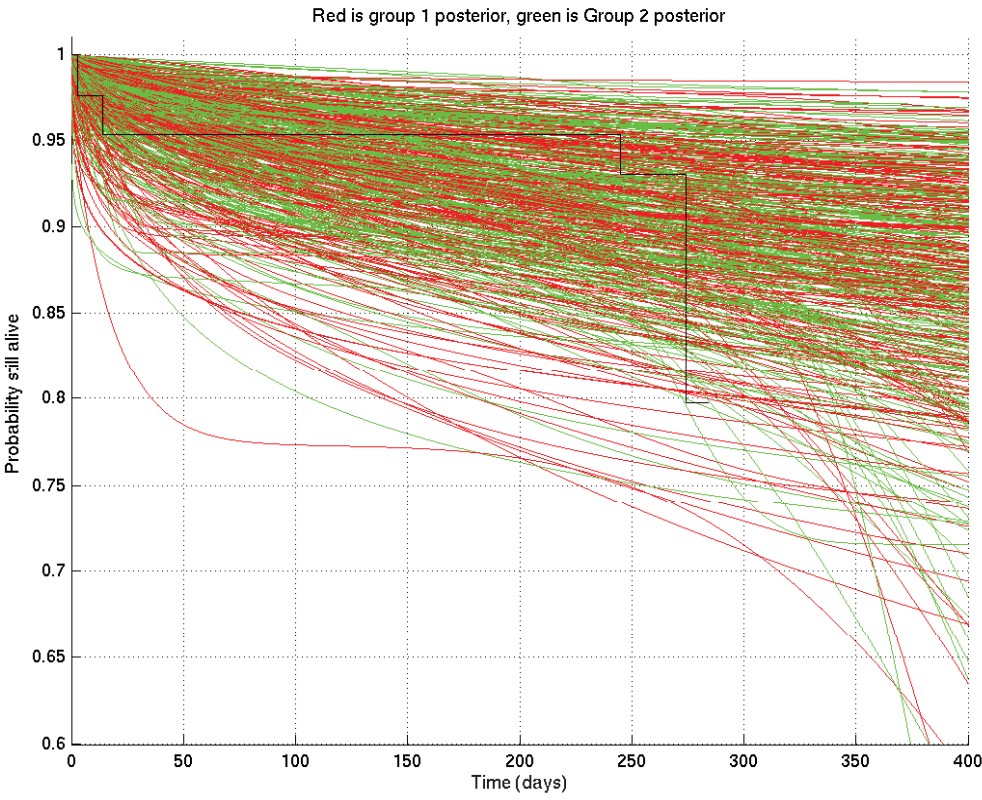

**Appendix 2—figure 16.** Samples captured from two runs on the same data started from different random values of the parameters, illustrating that the resulting distributions are essentially identical.

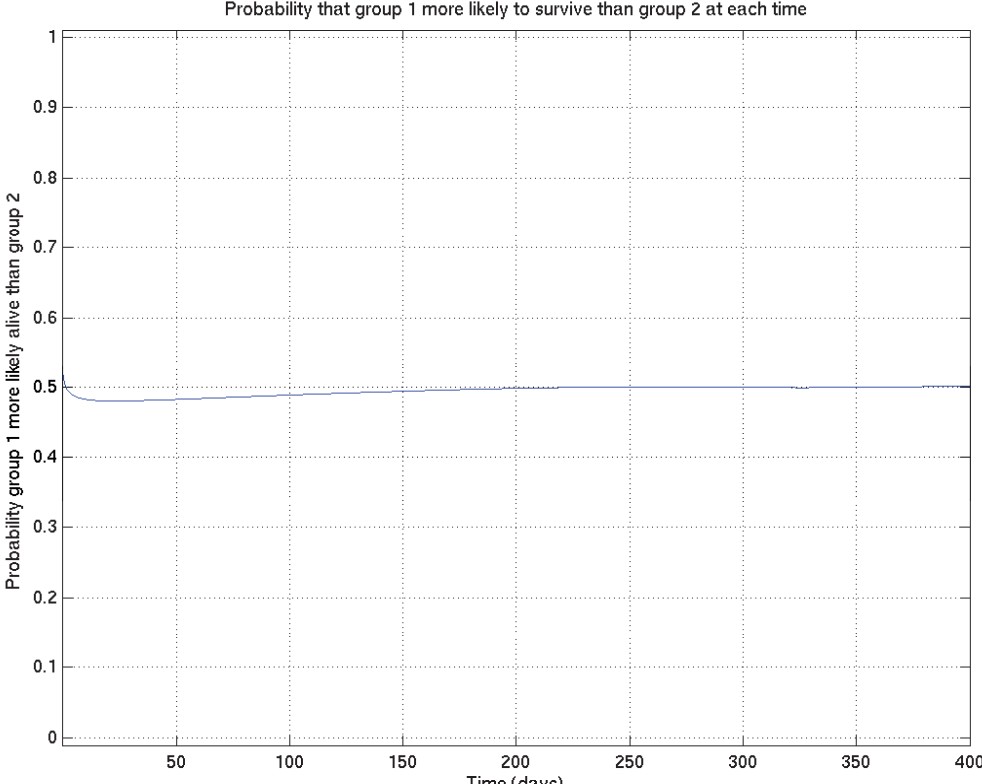

**Appendix 2—figure 17.** Comparison probabilities (analogous to *Appendix 2—figure 6*) for survival probability against time from two runs on the same data (and the same priors) started from different random values of the parameters. If the two distributions are identical (as they should be up to uncertainty caused by the non-infinite number of samples drawn during the MCMC runs), then at each time the probability that the 'red' distribution is greater than the 'green' (see *Appendix 2—figure 16*) should be 0.5 . Thus this plot, together with *Appendix 2—figure 16*, shows that the two distributions are essentially identical, and that the runs have converged to a common distribution.

## 4. Example of inference

Because, in the Results section of this paper, one particular specific example of Bayesian inference occurs whose interpretation is slightly tricky, it seems appropriate to discuss it specifically here. This corresponds precisely to the comparison of TT and non-TT genotypes in Grade 1 Indonesia patients.

We refer to *Appendix 2—figure 18*. The situation before knowledge of the data is described by the prior mean survival probability curve in solid magenta and its 2.5% and 97.5% centiles in dot-dash magenta, constructing the 95% prior confidence interval. (You may think, in the light of the data, that this prior is too pessimistic — but the point is that this is what was thought before knowledge of the data.) Note that the only parts of the plot outside this confidence interval are a small piece at the bottom left and an extremely thin sliver along the right-hand part of the top edge.

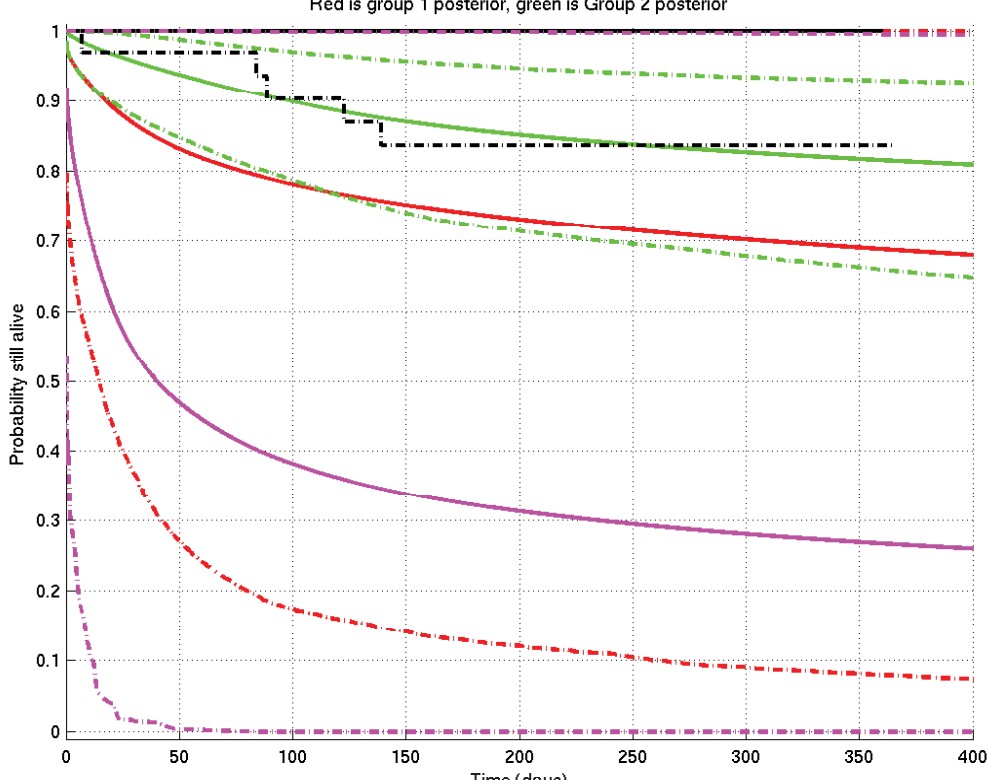

**Appendix 2—figure 18.** Example of inference whose interpretation is explained in detail in section 4 of this document. See also *Appendix 2—figure 19*. Prior mean and 2.5% and 97.5% centiles in magenta; posterior TT mean and centiles in red; posterior non-TT mean and centiles in green; Kaplan-Meier plots for TT in solid black and for non-TT in dash-dot black. There are 1 TT patient and 33 non-TT patients. (The Kaplan-Meier plot for group 1 (TT) and the upper centile plots for both prior and group 1 posterior are approximately coincident along the top of the graph).

We now collect the TT patients' data: there is, however, only 1 TT patient, who survives until 1 year before being censored. A single patient, however, has only a small effect on the prior (just as a single head-toss would not convince you a coin was biased): this shifts the posterior for the TT group upwards to the red lines, mean (solid) and centiles (dot-dash).

On the other hand when we collect the non-TT patients' data, there are 33 of them, so they have a bigger effect, both raising the mean and narrowing the 95% posterior confidence interval to the corresponding green plots. Even though these 33 patients survive less well than the single TT patient, they lift the posterior mean more than does the single TT patient, but the green 95% posterior confidence interval is much narrower than the red one.

Finally, analogous to *Appendix 2—figures 4* and *6*, we can calculate the probability that the TT population survives better than the non-TT population at each time point, getting *Appendix 2—figure 19*: the conclusion is that it has become very slightly less probable that TT survives better than non-TT than it was before (before it was 0.5 precisely), but in essence the posterior probability that TT survives better than non-TT remains not far off 0.5 throughout the time-course.

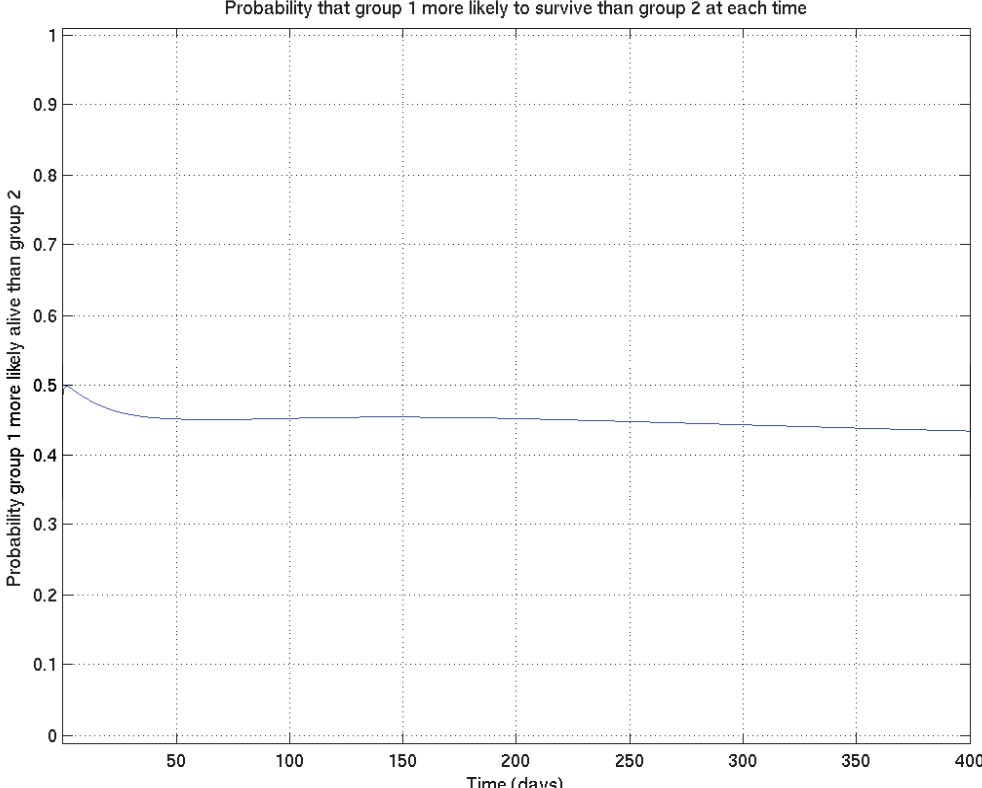

**Appendix 2—figure 19.** Example of comparison probabilities for inference whose interpretation is explained in detail in section 4 of this document. See also *Appendix 2—figure 18*.

Of course, in most examples in the paper, there are more patients in both groups being compared, and we are more likely to get a more definite conclusion.

## 5. Sensitivity to choice of priors

Specifically for comparisons of TT and nonTT subsets, where the subsets consist of very different numbers of patients, there is particular scope for otherwise unexpected sensitivity to the choice of uninformative priors. To assess this we initially checked, for one such comparison, the effect of using a different prior, namely:

$$a_J = 0.8$$

$$a_p = b_p = 1$$

$$a_m = b_m = 1$$

$$m_r = 0.5$$

$$r_r = 30 \text{ days}$$

$$N_k = 2$$

$$a_k = 1$$

$$\mathbf{b}_k = (1, 2)$$

$$\mathbf{c}_k = (1, 2),$$

which results in the set of samples of survival probability curves shown in *Appendix 2—figure 20*.

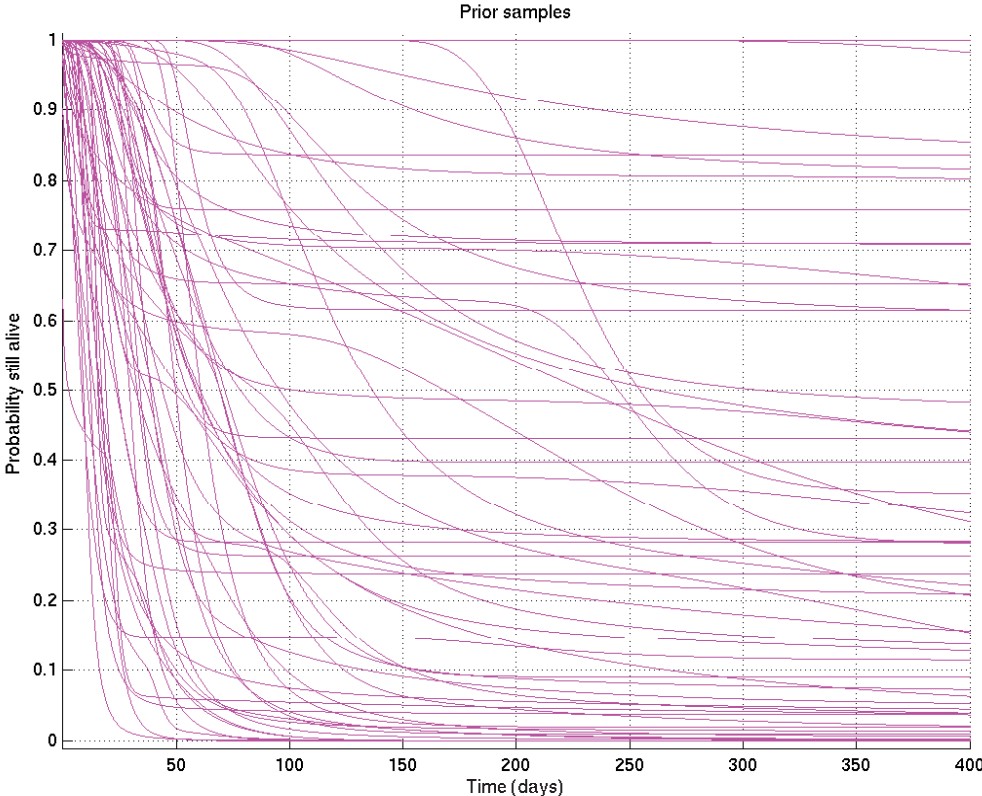

**Appendix 2—figure 20.** Samples of the survival probability against time for an alternative prior.

As can be seen by comparing this with *Appendix 2—figure 12*, this alternative prior does not envisage nearly as many early deaths in the first few days as the chosen prior, but equally believes it to be more unlikely that survival would be near 100% at much later times.

The effect of this on a comparison of a small subset and a large subset would be expected to be to shift the posterior on the small subset upwards in the early period and downwards in the late period compared with the large subset, increasing the significance of the early comparison if the small subset survived better than the large subset at that time, and reducing it if in the other direction (and vice versa at late times).

In the case of the comparison shown in *Appendix 2—figure 6* above, switching to the alternative prior gives *Appendix 2—figure 21*, indeed confirming this expectation: the peak significance, at around 10 days, increases from 0.994 to 0.996, while at 1 year the comparison probability reduces from 0.893 to 0.886 .

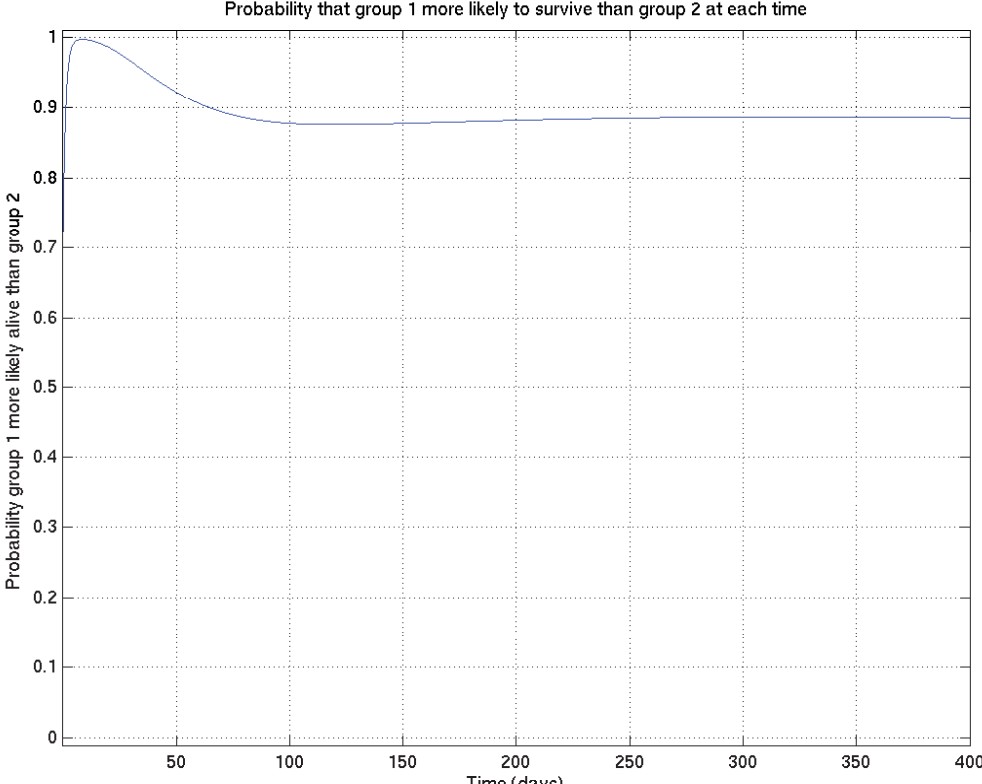

**Appendix 2—figure 21.** As for *Appendix 2—figure 6*, showing samples of the survival probability against time for an alternative prior.

We remark, however, that even with this significant change in the prior, the inferred comparison probabilities change remarkably little. We have therefore not reported detailed comparisons of alternative priors throughout the results.

