## [Decision Letter]

**Acceptance summary:**

Your description of the potential role of the LTA4H TT variant of Leukotriene A4 Hydrolase in mediating outcomes of tuberculosis meningitis has important implications for developing precision medicine approaches for this and other related inflammatory diseases. The use of Bayesian analysis in your study allowed for a detailed investigation of the contribution of this genotypic variant and has also pointed to other factors that may mediate treatment and disease outcomes, thus creating potential new avenues of research.

**Decision letter after peer review:**

Thank you for submitting your article "A Bayesian analysis of the association between Leukotriene A4 Hydrolase genotype and survival in tuberculous meningitis" for consideration by *eLife*. Your article has been reviewed by Bavesh Kana as the Senior Editor, a Reviewing Editor, and three reviewers. The reviewers have opted to remain anonymous.

The reviewers have discussed the reviews with one another and the Reviewing Editor has drafted this decision to help you prepare a revised submission.

As the editors have judged that your manuscript is of interest, but as described below that additional analyses are required before it is published, we would like to draw your attention to changes in our revision policy that we have made in response to COVID-19 (https://elifesciences.org/articles/57162). First, because many researchers have temporarily lost access to the labs, we will give authors as much time as they need to submit revised manuscripts. We are also offering, if you choose, to post the manuscript to bioRxiv (if it is not already there) along with this decision letter and a formal designation that the manuscript is "in revision at *eLife*". Please let us know if you would like to pursue this option. (If your work is more suitable for medRxiv, you will need to post the preprint yourself, as the mechanisms for us to do so are still in development.)

Summary:

Two Tuberculosis meningitis (TBM) studies conducted in Vietnam and Indonesia to assess the role of polymorphisms in the leukotriene A4 hydrolase (LTA4H) gene, implicated in modulating inflammation, yielded divergent conclusions. The submission by Whitman and colleagues utilizes a Bayesian approach to further investigate the data associated with these studies in an attempt to reconcile the observations. The underlying premise is that the Indonesian cohort was skewed towards more severe disease at presentation, which nullified the effect of the LTA4H genotype with patient survival in the Indonesian cohort.

1) By using a Bayesian approach in the analysis of both previous studies the authors are able to reconcile, in part, the divergent findings of an impact of LTA4H polymorphisms on TBM survival.

2) An unexpected finding was the protective effect of the LTA4H TT genotype on TBM response to glucocorticoid therapy in Vietnamese grade 3 cases. In contrast, in the Indonesian study the impact was found for grade 2 but not grade 3 patients.

3) The lack of effect by the TT genotype in the Indonesian Grade 3 TBM patients could be, as shown in the discussion, due to individuals who are genetically hyper susceptible to uncontrolled inflammation independently of LTA4H, differences in treatment and/or access to top-quality care.

The authors propose that LTA4H genotyping together with data on disease severity could be used to identify TB meningitis patients most likely to benefit from adjunctive glucocorticoid treatment. This is an important study and could shed light on the discordant results obtained from the studies conducted in Vietnam and Indonesia.

Essential revisions:

1) The authors state that the higher mortality in Indonesia was driven by a higher percentage of patients with Grade 2 disease (% with Grade 3 disease being equal in the two countries). They state that the LTA4H TT genotype effect does not extend beyond Grade 2 in Indonesia. In Vietnam though, the effect of TT is most pronounced in Grade 3. In people living with HIV, the TT effect was present in people with Grade 1 or 3 disease, but not with Grade 2 disease. The latter is dismissed as “likely spurious”. However, another explanation is that disparate results in different countries, different stages of disease, and different populations suggests that LTA4H is less useful than other factors (other genetics, environment, etc) in determining disease severity and possibility of treatment response with steroids. Can the authors provide a convincing argument otherwise?

2) Further to the point above, the MRC grade is generally thought to have good association with outcomes and thus, the similar mortality for Indonesia Grade 2 non-TT patients and Vietnam Grade 3 non-TT patients clearly suggests that grade and LTA4H genotyping may not always be able to identify TB meningitis patients most likely to benefit from adjunctive glucocorticoid treatment. Therefore, this conclusion should be toned down in the abstract and the manuscript. Perhaps LTA4H genotyping and mortality risk for each MRC grade (local to that setting) is most correlative of who is most likely to benefit from adjunctive glucocorticoid treatment? Importantly, it is still not clear why a grade-specific mortality difference was noted between the two cohorts. Was it related to the difference in critical / hospital care for these cohorts? Or other aspects? The authors speculate this but do not provide sufficient data, please address this.

3) In Indonesia, the average time to death was 8 days, and in Vietnam it was 50 days. The authors describe LTA4H TT effect as occurring over short or long periods of time, so this adds the element of longitudinal assessment and resulting effect to the mix. What effect would LTA4H play late in disease, once a patient is already on multidrug therapy as well as steroids, from a mechanistic point of view?

4) Was any analysis performed to test whether the patients removed from the cohorts did not skew the population? That is, were the demographic characteristics of the patients excluded from this analysis similar to those included in the study?

5) Provide p values for comparisons across the three sub-populations in Table 1 and Table 2 as well as Supplementary file 1?

6) While substantial details are presented in supplementary methods, a brief introduction (in simple language) on Bayesian methods would be useful for the reader to better understand the main methodology utilized in this study. For example, the statement in subsection “LTA4H TT genotype association with survival becomes stronger with increasing disease severity in Vietnam HIV-negative patients”, "Importantly, the model and priors used allowed us to incorporate our pre-existing knowledge that mortality risk to a population of TBM patients varies smoothly with time, rather than occurring at a number of discrete times common to all patients as is implied by the maximum likelihood solution illustrated by a Kaplan-Meier plot." helps understand the basis for the use of Bayesian analysis.

7) In most clinical trials, assumptions, sample size calculations and outcome measures are predefined to limit bias related to post-hoc analysis. Can the authors present some information on how bias was prevented? If this was not feasible, this should be listed as a study limitation.

8) These data are insightful, but as state above, it is still possible that another genotype or local characteristic of the population (Vietnam versus Indonesia) is / are confounding the results. For example, one of the co-authors of the current study has demonstrated that cerebral tryptophan metabolism (under strong genetic influence) is important for the outcome of TB meningitis (Lancet Infect Dis. 2018). Can the authors analyse the contribution other genetic factors based on currently available genotypic information from these cohorts or from other studies?

9) The authors suggest that “LTA4H genotyping together with disease severity assessment may target glucocorticoid therapy to patients most likely to benefit from it,” but they provide no road map for operationalising the finding. In whom should this genotyping be done? What role does the test result have in decisions regarding provision of steroids? Please provide a decision framework or a tool for use of this test plus data that a clinician would have.

[Editors' note: further revisions were suggested prior to acceptance, as described below.]

Thank you for resubmitting your work entitled "A Bayesian analysis of the association between Leukotriene A4 Hydrolase genotype and survival in tuberculous meningitis" for further consideration by *eLife*. Your revised article has been evaluated by Bavesh Kana (Senior Editor) and a Reviewing Editor.

The manuscript has been improved but there are some remaining issues that need to be addressed before acceptance, as outlined below:

Reviewers have accepted all revisions and explanations but request the following modifications:

You reword your conclusions to discuss additional limitations. For example, the statement, "Thus, LTA4H TT efficacy was limited by other factors that cause mortality. These factors appear independent of severity grade on presentation, and if they exceed a threshold (represented by about ~ 40% mortality) then the beneficial effect of LTA4H TT is lost." This is largely speculative and lacking in detail. What are these proposed factors? Carefully couching the conclusions in context of all limitations will enhance the value of your work and appeal to a broader audience who may be interested in these additional factors.

---

## [Author Response]

Essential revisions:1) The authors state that the higher mortality in Indonesia was driven by a higher percentage of patients with Grade 2 disease (% with Grade 3 disease being equal in the two countries). They state that the LTA4H TT genotype effect does not extend beyond Grade 2 in Indonesia. In Vietnam though, the effect of TT is most pronounced in Grade 3. In people living with HIV, the TT effect was present in people with Grade 1 or 3 disease, but not with Grade 2 disease. The latter is dismissed as “likely spurious”. However, another explanation is that disparate results in different countries, different stages of disease, and different populations suggests that LTA4H is less useful than other factors (other genetics, environment, etc) in determining disease severity and possibility of treatment response with steroids. Can the authors provide a convincing argument otherwise?

First, we have decided to remove the HIV cohort data from the paper so as to not distract from the core finding of the paper, namely that the use of Bayesian methods has enabled us to see that the LTA4H TT genotype does improve survival in the context of dexamethasone in Indonesia also, and that this effect is affected by mortality driven by other factors. An ongoing dexamethasone trial in HIV-positive patients who will be assessed for *LTA4H* genotype and other detailed parameters should resolve the question of whether HIV-positive individuals fare better with dexamethasone and if so, whether the TT genotype is relevant to their dexamethasone-responsiveness (Donovan et al., (2018).). We will use these analyses in that paper. The analytical methods and computational programs developed here will be invaluable in analyzing the results from that trial.

Second, we have no data on whether the *LTA4H* TT genotype determines disease severity and have not claimed that it does.

Finally, we respectfully differ from the reviewers’ suggestion that *LTA4H* is less useful than other factors in determining treatment response with steroids. Rather our careful and detailed analysis strongly suggests that the TT genotype does play a substantial and likely dominant role in steroid responsiveness, unless the patient is so sick upon entry with such advanced pathophysiology that they cannot be rescued by dexamethasone.

2) Further to the point above, the MRC grade is generally thought to have good association with outcomes and thus, the similar mortality for Indonesia Grade 2 non-TT patients and Vietnam Grade 3 non-TT patients clearly suggests that grade and LTA4H genotyping may not always be able to identify TB meningitis patients most likely to benefit from adjunctive glucocorticoid treatment. Therefore, this conclusion should be toned down in the abstract and the manuscript. Perhaps LTA4H genotyping and mortality risk for each MRC grade (local to that setting) is most correlative of who is most likely to benefit from adjunctive glucocorticoid treatment? Importantly, it is still not clear why a grade-specific mortality difference was noted between the two cohorts. Was it related to the difference in critical / hospital care for these cohorts? Or other aspects? The authors speculate this but do not provide sufficient data, please address this.

We are puzzled by the reviewers’ reasoning that “… the similar mortality for Indonesia Grade 2 non-TT patients and Vietnam Grade 3 non-TT patients clearly suggests that grade and LTA4H genotyping may not always be able to identify TB meningitis patients most likely to benefit from adjunctive glucocorticoid treatment.” In both cohorts, there is a clear association between MRC grade and survival with the Indonesia cohort faring worse at every grade. This finding came as a surprise to all of us because we had all thought that the difference in overall survival between the cohorts was simply because Indonesia had more severe Grade patients. This surprising result first became apparent when we analysed the TT versus non-TT data in Figure 2. To validate this, we performed the Grade-specific survival analyses in Figure 3 and then the direct head-on comparison of grade-specific survival in the two cohorts in Figure 4. If we take all these results together, the most parsimonious conclusion is that the TT genotype provides a benefit up to a point of no return, i.e. up to Grade 2 in Indonesia. After that it cannot help, for the reasons described in the paper and reiterated in the following paragraph. It is not surprising therefore, that in the absence of the TT genotype, Indonesia Grade 2 patients have a similar mortality to Grade 3 Vietnam patients, just as they do in the overall cohort (Figure 3).

In response to the reviewers’ further queries, we have ruled out that there was a hidden increased severity in Grade 3 Indonesia as reflected by the GCS scores which are sensitive for smaller changes in severity in Grades 2 and 3. While Grade 2 disease in Indonesia was somewhat more severe in Vietnam, Grade 3 disease was less severe. These new analyses are presented in Supplementary file 1 and in the Discussion.

Furthermore, we don’t think that we have downplayed the genotype-independent mortality factors. Rather, we have discussed them as extensively as we can in the Discussion. The Abstract also explicitly specifies that genotype-based therapy has its limitations: “However, its benefit is nullified in the most severe cases where other factors cause early mortality.” Based on our understanding of the levels of care available in Indonesia and Vietnam our suspicion is that some or most of these differences are attributable to the ability to provide expert intensive respiratory and other critical care support at the study site hospitals. We know that all patients in Vietnam were cared for in hospitals with critical care expertise whereas in Indonesia this was only possible for a small fraction of the cohort. However, this is a hypothesis that can only be addressed by patient-chart level review and by determination of the mortality rates for other critical illnesses at the same participating hospitals. We do not have those data, and thus have to leave this as an unproven hypothesis. Therefore, all that we can say is what is already in the Discussion: “The more likely possibility is that better ancillary care was possible in Vietnam where all patients were enrolled into a clinical trial versus only 17% in Indonesia (Thuong et al., 2017; van Laarhoven et al., 2017). Optimized respiratory support, in particular, would be essential to keep patients alive through the early high-risk stage in order allow for anti-inflammatory effects of corticosteroids to benefit the TT patients.”

3) In Indonesia, the average time to death was 8 days, and in Vietnam it was 50 days. The authors describe LTA4H TT effect as occurring over short or long periods of time, so this adds the element of longitudinal assessment and resulting effect to the mix. What effect would LTA4H play late in disease, once a patient is already on multidrug therapy as well as steroids, from a mechanistic point of view?

As shown in Figure 2, the *LTA4H* TT survival benefit starts early, with hazard rate ratios peaking in the range of 1-7 days. The finding that the biggest effect of dexamethasone on this genotype occurs early is not surprising given its role in countering inflammation. The reviewers are right – the effect wanes only gradually and in most cases does not vanish for several months or even through the analysis period. This is reflected in the TT-non-TT survival gap gradually increasing through this period in almost all cases. We think that this is more likely because of a knock-on benefit of the early reduction of acute inflammation by steroids rather than a different effect of the TT genotype in distinct pathophysiology operant only later. The former model would be more consistent with our understanding of TBM pathogenesis.

4) Was any analysis performed to test whether the patients removed from the cohorts did not skew the population? That is, were the demographic characteristics of the patients excluded from this analysis similar to those included in the study?

This information is now included in Supplementary file 1. We have decided that a formal analysis is not appropriate given that the reason that the patients were excluded is because they had key information missing. We knew the ages of all the patients who were excluded and the Bayesian comparison probability of the mean of the excluded distribution being greater than the included on was not significant (P 0.85, inverse-Γ preferred distribution).

5) Provide p values for comparisons across the three sub-populations in Table 1 and Table 2 as well as Supplementary file 1?

Bayesian comparison probabilities have now been computed for these comparisons for each line of Table 1 and Table 2 and described in the Materials and methods. The original Supplementary file 1, which pertained to the HIV-positive patients has been removed. (The new Supplementary file 1 contains the information described in the point 4 response above.)

6) While substantial details are presented in supplementary methods, a brief introduction (in simple language) on Bayesian methods would be useful for the reader to better understand the main methodology utilized in this study. For example, the statement in subsection “LTA4H TT genotype association with survival becomes stronger with increasing disease severity in Vietnam HIV-negative patients”, "Importantly, the model and priors used allowed us to incorporate our pre-existing knowledge that mortality risk to a population of TBM patients varies smoothly with time, rather than occurring at a number of discrete times common to all patients as is implied by the maximum likelihood solution illustrated by a Kaplan-Meier plot." helps understand the basis for the use of Bayesian analysis.

Thank you for encouraging us to describe this methodology further. We have added this information in Appendix 1, comparing frequentist and Bayesian paradigms and refer to it at the end of the paragraph discussing Bayesian analysis (Introduction).

7) In most clinical trials, assumptions, sample size calculations and outcome measures are predefined to limit bias related to post-hoc analysis. Can the authors present some information on how bias was prevented? If this was not feasible, this should be listed as a study limitation.

A major strength of Bayesian analysis is that bias due to post-hoc subgroup analysis doesn't arise, unless one only provides a selected subset of the results that suit one’s thesis. In response to the reviewers’ comment, we have included in the Introduction the following sentence:

“Finally, relevant to this re-analysis of completed clinical studies, Bayesian paradigms have less potential for bias arising from post-hoc analysis (Appendix 2).” We have also included details of this aspect of Bayesian analysis in Appendix 2 section 1.3. Since, we have now decided to withhold the HIV-positive data for the time being, we note that in this section, we have mentioned in it that we left out the HIV-positive data with the following lines:

“In this paper, the only important omission is that analysis was also done of the

HIV-positive patients in Vietnam, and this was not reported in this paper. This was because, although it tended to confirm similar findings to those from the HIV-negative cohort, there were some points that were less clear cut and harder to interpret, and its publication therefore awaits completion of a clinical trial in which the benefit of dexamethasone is being examined for HIV-positive patients of all three genotypes by randomizing them to get dexamethasone or not.”

8) These data are insightful, but as state above, it is still possible that another genotype or local characteristic of the population (Vietnam versus Indonesia) is / are confounding the results. For example, one of the co-authors of the current study has demonstrated that cerebral tryptophan metabolism (under strong genetic influence) is important for the outcome of TB meningitis (Lancet Infect Dis. 2018 May;18(5):526-535). Can the authors analyse the contribution other genetic factors based on currently available genotypic information from these cohorts or from other studies?

This is a good idea but beyond the scope of this study which was focused on the Indonesia *LTA4H* conundrum. It is our hope that the powerful analytical methods that we have developed here will be used by the community, and we are of course providing the code and methods with a commitment to help answer questions as possible.

9) The authors suggest that “LTA4H genotyping together with disease severity assessment may target glucocorticoid therapy to patients most likely to benefit from it,” but they provide no road map for operationalising the finding. In whom should this genotyping be done? What role does the test result have in decisions regarding provision of steroids? Please provide a decision framework or a tool for use of this test plus data that a clinician would have.

This is an interesting idea. Upon reflection, however, we think it is premature to suggest a decision algorithm until further information becomes available. In particular, we await the results of the ongoing trial randomizing CC and CT patients to dexamethasone or placebo. As with the HIV-positive patients, he analysis methods and computational programs developed for this study will be invaluable in analyzing the results of this trial.

[Editors' note: further revisions were suggested prior to acceptance, as described below.]

The manuscript has been improved but there are some remaining issues that need to be addressed before acceptance, as outlined below:Reviewers have accepted all revisions and explanations but request the following modifications:You reword your conclusions to discuss additional limitations. For example, the statement, "Thus, LTA4H TT efficacy was limited by other factors that cause mortality. These factors appear independent of severity grade on presentation, and if they exceed a threshold (represented by about ~ 40% mortality) then the beneficial effect of LTA4H TT is lost." This is largely speculative and lacking in detail. What are these proposed factors? Carefully couching the conclusions in context of all limitations will enhance the value of your work and appeal to a broader audience who may be interested in these additional factors.

We thank the reviewers for their careful review and are very pleased that they have approved our revisions.

We have made the following changes in response to their one outstanding request (Discussion section):

1) We have explicitly stated that we do not know for certain what the cause of excess Grade 3 LTA4H deaths in Indonesia is.

2) We have more clearly divided up the possible causes into non-genetic, ancillary care related causes and *LTA4H-*independent other genetic causes.

We hope this meets with the reviewers’ approval. The changes are here for the reviewers’ convenience:

“Thus the beneficial effect of dexamethasone to the *LTA4H* TT group may simply not have had time to come into play in Grade 3 Indonesia patients. A second possibility is that Indonesia Grade 3 patients presented with a greater degree of dysregulated inflammation in a manner not revealed by standard metrics of judging disease severity. If this were the case then corticosteroids might no longer be beneficial. Since both TT and non-TT patients suffered identical excess mortality risk, its cause would have been *LTA4H*-independent. Indonesia patients tended to be younger than Vietnam patients in all grades (Table 1), and perhaps were more prone to develop such a response. Such a previously unrecognized form of dysregulated inflammation might be caused by other genetic variants uniquely present in the Indonesia patients and would work through an inflammatory pathway that is less responsive to dexamethasone.”